# Assessment of isoprene and near surface ozone sensitivities to water stress over the Euro-Mediterranean region

Susanna Strada[1], Andrea Pozzer[2, 3], Graziano Giuliani[1], Erika Coppola[1], Fabien Solmon[4],
Xiaoyan Jiang[5], Alex Guenther[5], Efstratios Bourtsoukidis[3], Dominique Serça[4], Jonathan Williams[2, 3],
and Filippo Giorgi[1]

[1]The Abdus Salam International Centre for Theoretical Physics, Trieste (Italy)
[2]Max-Planck-Institut für Chemie, Mainz, Rheinland-Pfalz (Germany)
[3]Climate and Atmosphere Research Center, The Cyprus Institute, Nicosia, (Cyprus)
[4]Laboratoire d'Aèrologie, Toulouse (France)
[5]University of California (Irvine, USA)

**Correspondence:** Strada Susanna (sstrada@ictp.it)

**Abstract.** Plants emit biogenic volatile organic compounds (BVOCs) in response to changes in environmental conditions (e.g., temperature, radiation, soil moisture). In the large family of BVOCs, isoprene is by far the strongest emitted compounds and plays an important role in ozone chemistry, thus affecting both air quality and climate. In turn, climate change may alter isoprene emissions by increasing temperature as well as the occurrence and intensity of severe water stresses that alter plant

functioning.

The Model of Emissions of Gases and Aerosols from Nature (MEGAN) provides different parameterizations to account for the impact of water stress on isoprene emissions, which essentially reduces emissions in response to the effect of soil moisture deficit on plant productivity.

By applying the regional climate-chemistry model RegCM4chem coupled to the Community Land Model CLM4.5 and

10 MEGAN2.1, we thus performed sensitivity simulations to assess the effects of water stress on isoprene emissions and near-surface ozone levels over the Euro-Mediterranean region and across the drier/wetter summers over the period 1992–2016 using two different parametrizations of the impact of water stress implemented in the MEGAN model.

Over the Euro-Mediterranean region and across the simulated summers, water stress reduces isoprene emissions on average by nearly 6%. However, during the warmest and driest selected summers (e.g., 2003, 2010, 2015) and over large isoprene-

15 source area (e.g., the Balkans), decreases in isoprene emissions range from -20 to -60% and co-occur with negative anomalies in precipitation, soil moisture and plant productivity. Sustained decreases in isoprene emissions also occur after prolonged or repeated dry anomalies, as observed for the summers of 2010 and 2012. Although the decrease in isoprene emissions due to water stress may be important, it only reduce near-surface ozone levels by few percents due to a dominant NOx-limited regime over southern Europe and the Mediterranean Basin. Overall, over the selected analysis region, compared to the old

MEGAN parameterization, the new one leads to localized and 25–50% smaller decreases in isoprene emissions, and 3–8% smaller reduction in near-surface ozone levels.

# 1 Introduction

Plants release biogenic volatile organic compounds (BVOCs) in response to changes in biotic (e.g., insects, pathogens) and abiotic (e.g., temperature, water, sunlight) factors under both optimal and stressed conditions (Kesselmeier and Staudt, 1999; Niinemets, 2009). Among abiotic factors, air temperature, soil moisture, photosynthetic active radiation (PAR) and carbon dioxide ($CO_2$) strongly influence BVOC emissions (Peñuelas and Staudt, 2010). In the context of global climate change, it is critical to understand and characterize how BVOC emissions respond to changes in abiotic factors since these gaseous compounds influence the levels of greenhouse gases, aerosols and air pollutants, thus affecting both climate and air quality (Guenther et al., 1995; Pacifico et al., 2009). In turn, climate change may alter BVOC emissions by modifying, directly or indirectly, some of their drivers (e.g., proliferation of pathogens, global warming, rise in carbon dioxide levels) and by increasing the occurrence and intensity of severe stress (e.g., heatwaves, drought) (Peñuelas and Llusià, 2003; Pacifico et al., 2009; Peñuelas and Staudt, 2010), potentially exacerbating air pollution (Meleux et al., 2007; Colette et al., 2015; Churkina et al., 2017).

Isoprene dominates and accounts for at least 65% of the total BVOCs (Henrot et al., 2017). Once released in the atmosphere, it is rapidly oxidized and transformed into radical species that fuel reaction chains, linking its fate to the chemistry of the low troposphere and the boundary layer (Chameides et al., 1988; Atkinson and Arey, 2003; Pacifico et al., 2009; Laothawornkitkul et al., 2009). During daytime, under warm and sunny conditions, with high levels of nitrogen oxides (NOx), hydroxyl radical (OH) oxidizes isoprene to produce organic peroxy radicals that participate to the production of ground-surface ozone ($O_3$), a secondary air pollutant that threatens both human and plant health (Atkinson and Arey, 2003) and an important greenhouse gas when transported to the mid-troposphere.

The emission of isoprene may be altered by the effect of water stress on plant functioning (e.g., Bourtsoukidis et al., 2014). If high temperatures and radiation are well-known short-term drivers that enhance isoprene emissions up to an optimum, the effect of water stress is more variable. Reviews based on observational studies reported that severe/long-term water stress reduces BVOC emissions, whereas mild/short-term water stress temporarily amplifies or maintains BVOC emissions to protect plants against on-going stress (Niinemets, 2009; Peñuelas and Staudt, 2010; Werner et al., 2021; Byron et al., 2022). Regarding isoprene, recent meta-analysis of observational studies reported that isoprene emissions decrease by 23% when the relative water content drops to 55% (Feng et al., 2019) and show no intermediate increase under mild or short-term water stress (Bonn et al., 2019).

A parameterization of the decrease of isoprene emissions under water stress has been implemented in the BVOC emission model MEGAN (Model of Emissions of Gases and Aerosols from Nature, Guenther et al., 2006) and applied at global (Müller et al., 2008; Sindelarova et al., 2014; Zheng et al., 2015; Bauwens et al., 2016), regional (United States: Tawfik et al., 2012; Huang et al., 2014, 2015; Wang et al., 2017; Europe: Vogel and Elbern, 2021; Guion et al., 2022; Australia: Emmerson et al., 2019; China: Wang et al., 2021a, b) and local (Southern France: Genard-Zielinski et al., 2015; Texas: Seco et al., 2015) scales. These studies found that the original parameterization linking isoprene to water stress overly reduces emissions, with global decreases between -20% (Müller et al., 2008) and -50% (Sindelarova et al., 2014), reaching -70% over dry regions such as

Africa and Australia (Sindelarova et al., 2014; Bauwens et al., 2016). The scheme is also very sensitive to the modelling of soil moisture, plant rooting depth and of the soil wilting point (i.e., the minimal soil moisture below which plants can no longer draw water from the soil) (Müller et al., 2008; Genard-Zielinski et al., 2014; Huang et al., 2015; Seco et al., 2015; Wang et al., 2021b). Therefore, recently, the MEGAN parameterization was updated by Jiang et al. (2018) to account for the combined effect on isoprene emissions of soil moisture deficit and plant productivity (i.e., photosynthesis) under water stress. Using the the Earth System Model CAM-Chem-CLM4.5-MEGAN2.1, the authors found that the new parameterization reduces global isoprene emissions by $\sim$ 17%, with regional decreases of isoprene of up to 42% in dry areas. Applying the Earth System Model NASA GISS ModelE2.1, Klovenski et al. (2022) obtained a reduction of $\sim$ 3% in global isoprene emissions, while high-isoprene-emission regions such as the South-eastern U.S. showed larger decreases till -10% and -20%. Moreover, the authors suggested that the MEGAN soil moisture activity factor, and its parameters, should be tuned based on the modelling set-up, to which the soil moisture activity factor is sensitive.

Most of these studies apply MEGAN in its stand-alone version fed by meteorological reanalysis data (Müller et al., 2008; Genard-Zielinski et al., 2014; Sindelarova et al., 2014; Seco et al., 2015; Bauwens et al., 2016). Three studies, Jiang et al. (2018), Wang et al. (2021a), and (Guion et al., 2022) quantified the impact of changes in isoprene emissions due to water stress on atmospheric chemistry. At the global scale, Jiang et al. (2018) found contrasting effects on near-surface ozone, depending on the dominant photochemical regime: ozone increases in NOx-limited regimes (Amazon and Congo basins) while it decreases in VOC-limited regimes (Europe and North America). Over China, Wang et al. (2021a) simulated a decrease in near-surface ozone (-8%) and secondary organic aerosols (-30%). Over the Po Valley, (Guion et al., 2022) simulated a maximum decrease over land in near-surface ozone by -5%. Three studies, Genard-Zielinski et al. (2015), Vogel and Elbern (2021) and (Guion et al., 2022), investigated the effect of water stress on isoprene emissions over the Euro-Mediterranean region by applying the Guenther et al. (2006)'s parameterization over limited temporal (single summer) or spatial conditions (i.e., during field campaign in southern France, Genard-Zielinski et al., 2014, and in the Po Valley, Vogel and Elbern, 2021).

Compared to the Guenther et al. (2006)'s parameterization, the Jiang et al. (2018)'s parameterization does not simply include a soil moisture deficit effect, but it also accounts for vegetation processes, such as plant productivity. Recent observation-based studies demonstrated that plant productivity is the primary variable controlling plant water stress (Lee et al., 2013; Stocker et al., 2018; Pagán et al., 2019; Walther et al., 2019). Using multiple satellite-based proxies of plant productivity and soil moisture content from reanalyses, Walther et al. (2019) found that plant productivity strongly decreases when soil moisture is below the average over areas with limited or no tree cover (tree cover percentage lower than 50%), e.g. over southern France, Spain, the Balkans. Over these regions, Stocker et al. (2018) computed low-intermediate aridity indexes (between 0.3 and 0.9). Instead, in ecosystems where trees dominate (e.g., the Amazon and Congo Basin), plant productivity is more linked to co-variations in light availability and temperature than to water availability (Walther et al., 2019). Hence, depending on the ecosystem type, plant productivity has a different sensitivity to its main drivers, temperature, water and light availability.

When modelling isoprene emissions using MEGAN, the scientific community is still divided between including (Jiang et al., 2018; Emmerson et al., 2019) or not (Bauwens et al., 2018) the water stress effect on isoprene emissions. The need for this parameterization depends as well on the regional sensitivity to water stress. The Euro-Mediterranean region is a large

BVOC-source area, characterized by a warm-dry climate that regional climate projections indicate to become warmer and drier (Giorgi and Lionello, 2008; Giorgi et al., 2014), and where intense urbanization, air stagnation, and high temperatures often lead to harmful ozone pollution in summer (Meleux et al., 2007), with BVOCs contributing 30–75% to severe ozone pollution episodes (Solmon et al., 2004 and Curci et al., 2010 and references therein). From this brief overview, it is thus clear that more assessment is needed of the importance of water stress on isoprene emissions, particularly in regions prone to water stress. For this reason, here we focus on the Euro-Mediterranean region and we apply a regional climate-vegetation-chemistry model (the RegCM4, Giorgi et al., 2012) including MEGAN2.1 to quantify the effect of water stress on isoprene emissions and near-surface ozone levels, and to inter-compare the parameterizations of Guenther et al. (2006) and Jiang et al. (2018).

Differently to previous literature (e.g. Guion et al., 2022), numerical simulations of multi-year summers haven been used in this work, so to have statistically sound comparison of the model results of the influence of water stress on atmospheric chemistry, based on the different parameterizations present in the literature (Guenther et al., 2006; Jiang et al., 2018).

Our model and experiment design are described in the next section, while the results are presented in Section 3 and conclusions in Section 4.

## 2 Methodology

### 2.1 The regional climate model RegCM4 coupled to chem-CLM4.5-MEGAN2.1

For our experiments we use the RegCM4 limited-area model designed for long-term regional climate simulations (Giorgi et al., 2012). The model offers the flexibility to use either a hydrostatic or a non-hydrostatic dynamical core, and here we use the latter as described by Coppola et al. (2021). The model applies a $\sigma$-p vertical coordinate system run on a staggered Arakawa B-grid (i.e., velocities are evaluated at the grid center, masses at grid corners) and a relaxation exponential technique for lateral boundary conditions (Giorgi et al., 1993). We use the following model set-up: for radiative processes, the radiative transfer model from the Community Climate Model 3 (CCM3) of the National Center for Atmospheric Research (NCAR) (Kiehl et al., 1996); for convection, the parameterization of Tiedtke (1996) which accounts for sub-grid cloud heterogeneities to reproduce deep and shallow cumulus convection over land and sea; for the planet boundary layer (PBL) description, the University of Washington turbulence closure model (Grenier and Bretherton, 2001; Bretherton et al., 2004) where a convective mass flux scheme is coupled to a PBL turbulence scheme; for resolved-scale precipitation, the Subgrid Explicit Moisture scheme (SUBEX) that, based on the average grid cell relative humidity, distinguishes cloudy and non-cloudy fractions in each grid cell (Pal et al., 2000); for ocean fluxes (i.e., heat, freshwater, momentum), the bulk aerodynamic algorithm of Zeng et al. (1997); for land surface process, the Community Land Model (CLM version 4.5, Oleson et al., 2013; Sect. 2.1.1); for biogenic emissions, the Model of Emissions of Gases and Aerosols from Nature (MEGAN version 2.1, Guenther et al., 2012; Sect. 2.1.2). To represent chemical compounds and their reactions in the atmosphere, RegCM4 is coupled to the Global-Scale Carbon Bond Mechanism - Zaveri version (CBM-Z) gas-phase module (Zaveri and Peters, 1999; Shalaby et al., 2012; Sect. 2.1.3). Hereafter, we refer to the the surface-atmosphere model as RegCM4-CLM4.5, while RegCM4chem-CLM4.5-MEGAN2.1 is the surface-atmosphere-chemistry model.

### 2.1.1 The land surface model CLM4.5

To represent biophysical and biochemical processes linking the atmosphere to the land surface, the CLM4.5 land surface model solves the surface energy and water budget equations and computes stomatal physiology and photosynthesis (Oleson et al., 2013). In the present study, we do not activate the crop and urban models, nor the carbon and nitrogen cycle, and we apply static vegetation (i.e., no dynamic vegetation model).

     To maximize its ability to represent the large variety of vegetation species, the CLM4.5 model adopts the concept of plant
functional types (PFTs), which groups vegetation species sharing similar ecological and hydrological characteristics in a single PFT. The model includes 16 vegetated PFTs: eight for forests, three for shrub, three for grasslands and two for croplands. Bare soil represents an additional land-cover, for a total of 17 PFTs (Table S.1). The fraction cover of the 17 PFTs at a given grid-cell is prescribed referring to a present-day land-cover distribution from the Moderate Resolution Imaging Spectroradiometer (MODIS) and the Advanced Very High Resolution Radiometer (AVHRR) (Lawrence and Chase, 2007; Oleson et al., 2013).
When applying static vegetation, both vegetation structure (i.e., leaf and stem area indices, canopy top and bottom heights) and plant physiology are prescribed for each PFT using gridded data-sets (Lawrence and Chase, 2007; Bonan et al., 2002).

     The CLM4.5 model describes canopy, snow and soil hydrology. In particular, soil hydrology relies on a multi-layer module representing the vertical soil moisture transport across ten soil layers stretching from the surface down to $\sim 3$ m depth, with increasing soil thickness. The module for soil hydrology accounts for multiple processes along the vertical (i.e., infiltration,
surface and sub-surface runoff, gradient diffusion, gravity, canopy transpiration through root extraction, and interactions with groundwater), while horizontal exchange between soil water columns is neglected. Soil grid-cell points are initialized with a soil water content of $0.15$ m$^3$ m$^{-3}$ for all soil layers (Oleson et al., 2013). A five year model spin up is carried out to reach equilibrium in soil water content and initialize the whole water column.

     Soil hydrology and plant physiology are connected via the soil water stress function $\beta_t$ which directly limits stomata open-
ing (i.e., $\beta_t$ multiplies the minimum stomatal conductance) and indirectly reduces photosynthesis (i.e., $\beta_t$ multiplies leaf net photosynthesis and plant respiration) (Oleson et al., 2013). The soil water stress function ranges from 0 (dry soil) to 1 (wet soil) and, being lower than 1, it reduces stomatal conductance and photosynthesis. The function $\beta_t$ is dimensionless and depends on the root fraction distribution $r$ and the plant wilting factor $W$ in each soil layer $k$:

$$\beta_t = \sum_{k=1}^{Nb.\,soil\,lev} W_k\, r_k\,. \tag{1}$$

The root fraction distribution $r$ decreases exponentially with depth based on PFT-dependent parameters (see Table 8.3 in Oleson et al., 2013). The plant wilting factor $W$ is the minimal soil moisture below which plants cannot extract water from the soil. In each soil layer, $W_k$ depends on the relative porosity of the selected soil layer and the effective free energy that allows water to move in that layer:

$$W_k = \frac{\phi_c - \phi_k}{\phi_c - \phi_o}\left(\frac{\theta_{sat,k} - \theta_{ice,k}}{\theta_{sat,k}}\right), \tag{2}$$

where $\phi$ is the soil water matric potential (mm), which depends on the soil wetness (dimensionless); $\phi_c$ and $\phi_o$ are PFT-dependent parameters that define the soil water matric potential when stomata are fully closed or fully open, respectively; $\theta_{sat}$ is the total porosity; $\theta_{sat} - \theta_{ice}$ is the effective porosity accounting for the ice fraction.

### 2.1.2 The biogenic emission model MEGAN2.1

The biogenic emission model MEGAN describes BVOC foliage emissions via a mechanistic algorithm combining information on average emissions and their response to changes in environmental conditions. The model includes the emissions of 150 compounds, which are lumped in 19 emission categories to reduce computational costs (Guenther et al., 2012). Due to its abundance, isoprene represents a single class and a single compound.

For the compound class $i$, the net emission rate $F$ results from the multiplication of a compound- and PFT-dependent emission factor $\epsilon$ and a compound-dependent normalized empirical function $\gamma$ describing the dependence of emissions on environmental conditions, everything scaled by the grid-cell fraction covered by a specific PFT $\chi$:

$$F_i = \gamma_i \sum_{j=1}^{Nb.PFT} \epsilon_{i,j}\, \chi_j \,. \tag{3}$$

In the first version of MEGAN, BVOC emissions only depended on photosynthetically active radiation (PAR) and leaf temperature (Guenther et al., 1993). In MEGAN2.0, Guenther et al. (2006) included the dependence of BVOC emissions on other drivers in the compound-dependent activity emission factor $\gamma_i$:

$$\gamma_i = C_{CE} \times LAI \times \gamma_{PAR,i} \times \gamma_{T,i} \times \gamma_{age,i} \times \gamma_{SM,i} \times \gamma_{CO_2,i} \,. \tag{4}$$

In Equation 4, the canopy environment coefficient $C_{CE}$, which describes the canopy loss/production, multiplies the leaf area index (LAI) and the activity emission factors describing the effects of PAR, temperature, leaf age, soil moisture and $CO_2$ concentrations on BVOC emissions.

The soil moisture activity factor $\gamma_{SM}$ is only applied to isoprene and represents a reduction in isoprene emissions under severe water stress. Guenther et al. (2006) proposed a first version of $\gamma_{SM,2006}$ based depends on the root fraction in a given soil layer, the useful soil moisture compared to the soil moisture at the wilting point $\theta_w$, and a sensitivity parameter $\Delta\theta_1$ taken from a study for a single plant species (Pegoraro et al., 2004):

$$\gamma_{SM,2006} = \begin{cases} 0 & \theta < \theta_w \text{ severe water stress} \\ \frac{\theta - \theta_w}{\Delta\theta_1} & \theta_w < \theta < \theta_1 \\ 1 & \theta > \theta_1 \text{ no water stress} \end{cases} \tag{5}$$

Recently, Jiang et al. (2018) modified the MEGAN soil moisture activity factor to link isoprene emissions to drought response of photosynthesis (already parameterized in land surface models). In its new version, the soil moisture activity factor

$\gamma_{SM, 2018}$ depends on the soil water stress function (Eq. 1), the maximum rate of carboxylation by the photosynthetic enzyme Rubisco ($V_{cmax}$) and an empirical parameter $\alpha$ derived from field measurements (Potosnak et al., 2014; Seco et al., 2015):

$$\gamma_{SM, 2018} = \begin{cases} \frac{V_{cmax}}{\alpha} & \beta_t < 0.6, \alpha = 37 \\ 1 & \beta_t \geq 0.6 \text{ no water stress} \end{cases} \tag{6}$$

In CLM4.5, $V_{cmax}$ changes with leaf temperature and water stress. Specifically, the soil water stress function $\beta t$ multiplies $V_{cmax}$ and plant respiration, thus directly limiting $CO_2$ uptake and photosynthesis and indirectly influencing the stomatal conductance process.

### 2.1.3 The RegCM4 chemistry module CBM-Z

The CBM-Z module provides RegCM4 with 57 chemical species and 124 chemical reactions which include oxidation, dissociation and photolysis (Zaveri and Peters, 1999; Shalaby et al., 2012). RegCM passes input variables to the CBM-Z module, specifically: surface emissions, atmospheric and chemical boundary conditions, and fixed chemical concentrations for selected species (e.g, O2).

As emission input, we use anthropogenic emissions from the baseline simulation of the ECLIPSE project, version 5A (Stohl et al., 2015; Klimont et al., 2017, Evaluating the Climate and Air Quality Impacts of Short-Lived Pollutants,). Over the period 1990–2050, the ECLIPSE database provides monthly, global, gridded emissions of methane and a number of short-lived climate forcers (black and organic carbon, carbon monoxide, nitrogen oxides, tropospheric ozone, sulfur dioxide, ammonia, and non-methane volatile organic compounds). The database includes shipping emissions and a seasonal cycle for each species, while it does not include pyrogenic emissions. For this reason, we have combined ECLIPSE with the GFED4 inventory (Giglio et al., 2013, Global Fire Emissions Database fourth generation,) for the following species: black and organic carbon, carbon monoxide, nitrogen oxides, tropospheric ozone, and non-methane volatile organic compounds (NMVOCs). Over the period 1995–2016, GFED4 provides monthly fire emissions at a spatial resolution of $0.25^{circ}$ (van der Werf et al., 2017), based on emissions factors presented in the literature (Akagi et al., 2011; Andreae, 2019).

In both the ECLIPSE and the GFED4 databases, NMVOC emissions are lumped together. The CBM-Z module distinguishes 13 compound/class of NMVOCs, hence we have disaggregated NMVOC emissions into the CBM-Z compound/class using conversion factors from Szopa et al. (2005) and other sources. We have derived each compound/class emissions by multiplying NMVOC emissions by the respective fraction of total emitted carbon reported in Table 1. For lumped compounds (e.g., ACET, ketones, or PAR, paraffin carbon), the fraction of total emitted carbon results from the sum of fractions of single compounds taken from Zaveri and Peters (1999) and Szopa et al. (2005) (Table 2).

Chemical boundary conditions are taken from the 1999–2009 monthly climatology built on the global chemistry model Model for Ozone and Related chemical Tracers (MOZART) Emmons et al. (2010). The species O2 and H2 are kept at fixed concentrations (Graedel et al., 1993).

The RegCM model handles the removal of chemical species via dry and wet deposition, with dry deposition representing the main sink for for ozone. Dry deposition is parameterized for releavant species (including ozone) using the scheme of Zhang

et al. (2003) which reproduces stomatal and non-stomatal uptake from plants and soil by applying three resistances in series: aerodynamic, quasi-laminar and bulk surface/canopy resistances.

More details on the CBM-Z module and the RegCM4-chem model can be found in Zaveri and Peters (1999) and in Shalaby et al. (2012).

## 2.2   Simulation and analysis design

We carry out atmosphere-only simulations with the RegCM4CLM4.5 model, while we use the full model RegCM4chem-CLM4.5-MEGAN2.1 to run atmosphere-chemistry simulations. Both simulations include surface-atmosphere interactions,
handled by the CLM4.5 land surface model. All simulations are run at a horizontal grid spacing of  25 km over the Med-CORDEX domain (Ruti et al., 2016, http://wcrp-cordex.ipsl.jussieu.fr), forced with ERA-Interim reanalysis (Dee et al., 2011), which provides lateral and boundary conditions for both the atmosphere and the sea surface temperature every six hours at a horizontal resolution of $0.75°$. Greenhouse gas concentrations follow the historical values from the Inter-governmental Panel on Climate Change (IPCC) Fifth Assessment Report (AR5) (van Vuuren et al., 2011). Stratospheric ozone is prescribed based
on the data-set used for the Coupled Model Intercomparison Project (CMIP5) from the SPARC project (Stratospheric Processes and their Role in Climate) (Cionni et al., 2011), while aerosols are not accounted for. Total solar irradiance is taken from the CMIP5 solar forcing data (Lean et al., 1995).

Firstly, we perform a control atmosphere-only simulation (named "ATM"), covering the period 1987–2016 (with a 5-year spin-up, 1987–1991). These simulations provide initial and lateral meteorological and surface conditions to drive the coupled
atmosphere-chemistry simulations. This methodology ensures that at the first time step of each atmosphere-chemistry simulation atmospheric and soil conditions are realistic. The atmosphere-chemistry simulations are then carried out for 12 selected summers: 1992, 1994, 1995, 1997, 2000, 2003, 2006, 2007, 2010, 2012, 2014, 2015, since summer is the season of maximum isoprene emissions and maximum production of near-surface ozone. These summers have been selected based on the 1970–2016 precipitation climatology derived from the E-OBSv20 data-set as those having dry or wet anomalies within the
period 1992–2016 (Fig. S.1).

The summer atmosphere-chemistry simulations cover the periods May through August, with May assumed as spin-up to reach the equilibrium of chemical species (and thus not considered in the analysis). Each simulation is run twice: with ("GAMMA-SM2018on") and without ("GAMMA-SMoff") the new MEGAN soil moisture activity factor ($\gamma_{SM,2018}$, Eq. 6). We then compare the GAMMA-SM2018on and GAMMA-SMoff simulations to evaluate the effect of soil moisture on isoprene
emissions and near surface ozone levels. We also selected some extreme summers (i.e., 1994, 2003, 2010) to perform the same simulation using the old MEGAN soil moisture activity factor ($\gamma_{SM,2006}$, Eq. 5, "GAMMA-SM2006on") in order to compare the effects of the updates in the activity factor parameterization. To ensure there are no differences in the simulated climate across the atmosphere-chemistry simulations (GAMMA-SM2018on, GAMMA-SM2006on and GAMMA-SMoff), we do not account for the chemistry feedback on climate. Table 3 summarizes all performed simulations.

We focus our analysis on the summer atmosphere-chemistry simulations for the June-July-August (JJA) periods, when isoprene emissions and surface ozone levels are maximum over the Med-CORDEX domain. We compute the absolute differences

in a variable $X$ as $\Delta X = X_{GAMMA-SM*on} - X_{GAMMA-SMoff}$. Percentage changes in $X$ are calculated relative to the "GAMMA-SMoff" simulation (i.e., $\Delta_\% X = \Delta X / X_{GAMMA-SMoff} \times 100$). When considering the old and new MEGAN soil moisture activity factors, we compare the GAMMA-SM2018on and GAMMA-SM2006on simulations against each other and as well against the GAMMA-SMoff simulations.

We derive values of the soil moisture activity factors $\gamma_{SM}$ by dividing isoprene emissions from the GAMMA-SM2018on (or the GAMMA-SM2006on) simulation by emissions from the reference simulation GAMMA-SMoff.

Except for the model evaluation of atmospheric fields, which considers the whole Med-CORDEX domain, our analysis encompasses part of the domain, whose coordinates are [15°E–50°W; 28°–58°N]; hereafter, we refer to this area as the Euro-Mediterranean region. To facilitate the model evaluation, the model output was re-mapped onto the observed data grid using a distance-weighted method, which conserves both the total field and its spatial structure (Torma et al., 2015).

# 3   Results

## 3.1   Model evaluation

### 3.1.1   Atmospheric fields: temperature and precipitation

To evaluate mean and extremes in temperature and precipitation, we use the E-OBS ensemble version 20e (E-OBSv20e), which provides daily, land-only, station-based, gridded precipitation and surface air temperature (mean, minimum and maximum) over the period 1950–2018 at a resolution of $0.25°$ (Cornes et al., 2018). For cloud cover, we use the CLAAS (CLoud property dAtAset using SEVIRI, version 1) observation data-set produced by the European Organisation for the Exploitation of Meteorological Satellites (EUMETSAT) within the CM SAF project (Satellite Application Facility on Climate Monitoring), which is derived from the geostationary Meteosat Spinning Enhanced Visible and Infrared Imager (SEVIRI) measurements and provides global (between $65°$S and $65°$N), monthly mean, fractional cloud cover over the period 1991–2015 at a resolution of $0.05°$ (Stengel et al., 2014). Table S.2 presents all observation-based data-sets, and their features, that we used for model evaluation in the present study.

Figure 1 presents the spatial distribution of summer-averaged atmospheric biases for the period 1992–2016. Compared to E-OBSv20e, the RegCM4-CLM4.5 model shows a prevailing cold bias between $-3°$ and $-2°$ C across the Mediterranean Basin. Conversely, over south-western Russia, northwestern Africa and Iraq, there is a warm bias of $+1°$ to $+2°$ C (Fig. 1.a). Overall, the bias absolute values are mostly in the range of $1°$ to $3°$ C, which is in line with other regional climate simulations (Kotlarski et al., 2014; Vautard et al., 2021). For precipitation over land, RegCM4-CLM4.5 shows a prevailing wet bias across the domain in the range of $+1$ and $+4\ \mathrm{mm\,day^{-1}}$), with larger biases in the mountainous regions (i.e., the Pyrenees, the Alps, the Carpathians) (Fig. 1.b). Compared to the CLAAS data-set, the model tends to overestimate cloudiness (between 10 and 40% in terms of total cloud fraction) over the ocean and the Mediterranean Sea, as well as over northern Spain, southern France, northern Italy and northern Turkey (between 10 and 20%), while cloudiness is underestimated (between -5 and -20%) over the southern, eastern and northern parts of the domain (Fig. 1.c). In terms of climate extremes, compared to E-OBSv20e,

the summer bias in daily minimum temperatures shows both cold and warm bias, between $-3°$ to $+3°$ C (Fig. 1.d), while daily maximum temperatures show prevailing cold biases (between $-5°$ and $-1°$ C) over nearly the entire domain (Fig. 1.e). The model overestimates intense precipitation over most of the domain with a wet bias between $+10$ and $+15\,\mathrm{mm\,day^{-1}}$ in the 99th percentile of daily precipitation (Fig. 1.f). It is likely that the wet precipitation biases and the cold temperature ones are related, but overall the model performance for the present runs is in line with previous applications of the RegCM4 (e.g., Fantini et al., 2018).

### 3.1.2 Surface fields: latent heat flux and soil moisture

To evaluate surface latent heat flux as an indicator of evapotranspiration, we use the remote-sensed (RS) product from the FLUXCOM data-set (Jung et al., 2019), which provides global, monthly, gridded remote-sensed latent heat fluxes for 2001–2015 at a resolution of $0.5°$. The FLUXCOM data-set results from the upscaling of site-level energy flux observations from the global network of micro-meteorological flux measurement towers (FLUXNET eddy-covariance data) using a machine learning technique and combining in-situ observations with remote sensing and land cover data (Jung et al., 2019). Compared to FLUXCOM over the Euro-Mediterranean region, RegCM4-CLM4.5 shows larger evapotranspiration during summer, over land-only grid cells ($5$–$10\,\mathrm{W\,m}-2$, Fig. 2.a; $+10$–$20\%$, percentage biases in Fig. S.2), with the largest wet bias over north-eastern Spain ($+20$–$40\,\mathrm{W\,m^{-2}}$; $+40$–$80\%$), while some areas located in the middle and southern part of the domain (Portugal, north-western Spain, western France, middle of Italy, the Balkans, northern Africa and the Middle East) display a dry bias (between -5 and -20 W'm$^{-2}$, between -10 and -20%). From June to August, over land-only grid cells located in the southern part of the domain (below $50°$N), the difference between model and observation annual cycles is $< 5\,\mathrm{W\,m^{-2}}$ (Fig. 2.b). The overestimation of latent heat flux is likely related to the overestimation of precipitation found in Section 3.1.1.

For the evaluation of soil moisture, we use the global, monthly, gridded ($0.25°$) surface soil moisture (SSM) data for 1978–2015 produced by the European Space Agency Climate Change Initiative (ESA-CCI, version 04.4) based on observations from both scatterometers and radiometers (Dorigo et al., 2017). The ESA-CCI data-set offers three products obtained from different sensor-type : ACTIVE (scatterometer-only), PASSIVE (radiometer-only) and COMBINED (merged ACTIVE and PASSIVE products), and here we use the COMBINED product. Since the ESA-CCI data-set has many missing values over Europe before 2004, we focus our analysis on the period 2005–2015. The ESA-CCI theoretical global mean sensing depth is of $\sim 2\,\mathrm{cm}$ (Dorigo et al., 2017). Based on the soil thickness of the RegCM4 soil layers (Table S.3), we compare ESA-CCI data with the first model soil layer (soil depth: $1.75\,\mathrm{cm}$), which shows a similar pattern as the observed, with lower values in the eastern part of the domain (Fig. S.3). Over the southern part of the Euro-Mediterranean region (south of $50°$N), RegCM4-CLM4.5 follows well ESA-CCI volumetric soil moisture and shows high correlation ($r^2 = 0.7$), while the correlation is lower north of $50°$N ($r^2 = 0.3$), most probably due to the effect of snow at higher latitudes (Fig. 3.a and 3.b). South of $50°$, both RegCM4-CLM4.5 and ESA-CCI SSM show a bimodal distribution, which however peaks around lower values in RegCM4-CLM4.5 (below $0.1\,\mathrm{m^3\,m^{-3}}$ against 0.1 and $0.2\,\mathrm{m^3\,m^{-3}}$ for ESA-CCI SSM), and spans a larger range (up to $0.5\,\mathrm{m^3\,m^{-3}}$ vs. $0.4\,\mathrm{m^3}$ $\mathrm{m^{-3}}$ for ESA-CCI SSM) (Fig. 3.c and 3.d). Hence, compared to observations, the model reproduces more frequently very dry (below $0.1\,\mathrm{m^3\,m^{-3}}$) and very wet (above $0.4\,\mathrm{m^3\,m^{-3}}$) grid cells over the Euro-Mediterranean region.

Overall, compared to observations, RegCM4-CLM4.5 reproduces quite well fields of evapotranspiration and soil moisture over the Euro-Mediterranean region.

### 3.1.3 Chemical fields

Among the chemical species simulated by the RegCM4chem-CLM4.5-MEGAN2.1 model, we focus on isoprene, formaldehyde (HCHO) and near surface ozone, and we evaluate model output produced by the RegCM4chem-CLM4.5-MEGAN2.1 model in the GAMMA-SMoff simulation.

For isoprene, since there is no network over Europe routinely measuring isoprene concentrations in vegetated areas, we use in-situ mearurements of isoprene concentrations collected during two field-campaigns that took place in: (1) in southeastern France (site: La Verdière; Latitude: $43.63°$ N, Longitude: $5.93°$ E) during the summer 2000 (from June 21 to July 6) in the framework of the ESCOMPTE field campaign (Cros et al., 2004) when isoprene concentrations had been measured every 30 minutes using a Fast Isoprene Sensor; (2) in Cyprus (site: Ineia; Latitude: $34.96°$ N, Longitude: $32.39°$ E) during the summer 2014 (from July 7 to August 3; data collected every 45 minutes) using the technique of gas chromatography - mass spectrometry (GC-MS) (Derstroff et al., 2017). In general, the model underestimates isoprene concentrations at both locations (La Verdiére and Ineia) and sometimes it reproduces a delayed peak in isoprene concentrations compared to observations. Differences between observations and model output could result from multiple factors: (i) the cold and wet model bias that limits isoprene emissions; (ii) differences between vegetation types on the field and in the model grid-cell; and (iii) different scales with a model grid-cell spanning over a surface of hundreds of square kilometers, while station measurements have a footprint of a few hundreds of meters.

In addition, to evaluate isoprene emissions, we use satellite retrievals of formaldehyde, an intermediate by-product of the oxidation of hydrocarbons such as methane and BVOCs. Although the oxidation of methane represents the dominant source of formaldehyde (60%) followed by the oxidation of BVOCs (30%) (Stavrakou et al., 2015), methane contributes to the background abundance of formaldehyde in the troposphere (Fowler et al., 2009), while isoprene drives major variations of formaldehyde concentrations in the boundary layer, with contributions of up to 85% during the growing season (Franco et al., 2016) at a spatial scale of ca. 10–100 km (Palmer et al., 2003). Thus, retrievals of HCHO provides a good surrogate to evaluate the model performance in reproducing isoprene emissions.

To evaluate the formaldehyde burden, we use retrievals of HCHO column concentrations from the Ozone Monitoring Instrument (OMI), a nadir-viewing UV/Vis solar backscatter instrument, travelling aboard the Aura satellite (Stavrakou et al., 2018). These are gridded, Level 3, monthly mean data produced in the framework of the EU FP7 project Quality Assurance for Essential Climate Variables, version QA4ECV (De Smedt et al., 2015), covering the period 2005–2016 at a resolution of $0.25°$. Focusing on land-only grid-cells in the Euro-Mediterranean region, for the simulated summers within the OMI observing period (2005–2016), RegCM4chem-CLM4.5-MEGAN2.1 shows similar distributions of summer-averaged HCHO column concentrations as observed. However, the model has lower median values and a smaller variability in maximal outliers (Fig. 5). This result is consistent with the model underestimating isoprene concentrations compared to in-situ measurements. Summer 2010, when a heatwave hit Russia with associated wildfires (Barriopedro et al., 2011), stands out as the summer when the

model does not capture the largest median value of HCHO columns, although the model shows larger variability in outliers. These differences may be related to the cold and wet biases identified for the RegCM4-CLM4.5 model (Sect. 3.1.1) since warm atmospheric conditions favour BVOC production and emissions and, lastly, influence HCHO columns. Negative values in the OMI data-set indicate high noise in the data over areas with low HCHO columns, mainly located over northern Africa in the chosen domain.

To evaluate near surface ozone, we use air quality measurements collected by the European Environment Agency (EEA) that provide observations over different European countries (https://www.eea.europa.eu/data-and-maps/data/aqereporting-9). In Fig. 6 the summary of the comparison is presented, focusing on the summer 2015 and using daily means. Although RegCM4chem-CLM4.5-MEGAN2.1 undrestimates ozone concentrations compared to the EEA data-set over the three se-lected countries (i.e., Italy, Spain, and France) and the Airbase observations span till very low values (close to zero), observed and modelled ozone concentrations differ of less than $50\ \mu\mathrm{g\,m^{-3}}$ over Italy and less than $25\ \mu\mathrm{g\,m^{-3}}$ over Spain and France. Additional comparisons against the re-analyses from the Copernicus Atmosphere Monitoring Service (CAMS, Marécal et al., 2015) for the period 2003–2007 shows that the model performs well over land (differences lower than 10 ppbv), while it un-derestimates near-surface ozone over the Mediterranean Basin, with differences spanning between 10 and 20 ppbv and with some summers and some grid-cells showing larger differences, up to 30 ppbv (Fig. S.4).

## 3.2 Effect of water stress on isoprene emissions

In MEGAN 2.1, the activity factor $\gamma_{SM,2018}$ spans between 0 and 1 and, based on the intensity of water stress, it reduces isoprene emissions (Eq. 6). Figure 7 shows the spatial distribution of the unitless summer-averaged soil moisture activity factor $\gamma_{SM,2018}$ across the simulated summers and across areas where vegetation can grow (desert areas are in gray); while Figure 8 shows the spatial distribution of summer percentage changes in isoprene emissions (absolute changes in Fig. S.5). Low values of the soil moisture activity factor ($0 < \gamma_{SM,2018} < 0.2$) indicate a sustained water stress and are located over dry climates such as over northern Africa and the Middle East (Fig. 7). In these areas isoprene emissions decrease by more than 80% when the effect of water stress is accounted for in numerical simulations (Fig. 8). Mid/high values ($0.6 < \gamma_{SM,2018} < 1.0$) correspond to a mild/low water stress and are located over areas that experienced a dry anomaly based on the 1970–2016 precipitation climatology (Fig. S.1 vs. Fig. 7). Over these areas, isoprene emissions decrease between -5 and -60% (Fig. 8) (absolute changes between -0.50 and -12.0 $\mathrm{mg\,m^{-2}\,day^{-1}}$, Fig. S.5).

Simulated decreases in isoprene emissions are similar to those simulated over Europe across year 2010 by Jiang et al. (2018) using a global climate model who found a decrease in isoprene emissions over Europe with a maximum in August and spanning between 0 and -2 $\mathrm{mg\,m^{-2}\,hour^{-1}}$. In the summer 2010, the RegCM4chem-CLM4.5-MEGAN2.1 model also reproduces the largest decrease in isoprene emissions, with a maximum reduction of -76 $\mathrm{mg\,m^{-2}\,day^{-1}}$ simulated in July and located over south-western Russia (latitude: 60.24°N; longitude: 39.88°E) where drought and an extreme heat wave occurred (Barriopedro et al., 2011). Such a substantial reduction in isoprene emissions corresponds to -3 $\mathrm{mg\,m^{-2}\,hour^{-1}}$ (not shown). Compared to another high-isoprene-emission regions such as the South-eastern U.S., the Euro-Mediterranen region shows similar reductions in summer isoprene emissions, between -10% and -20% as simulated by Klovenski et al. (2022).

Table 4 presents absolute and percentage changes in isoprene emissions and near surface ozone mixing ratios averaged over the Euro-Mediterranean region. Over this region and across the simulated summers, isoprene emissions decrease by 6% on average (-0.3 $\mathrm{mg\,m^{-2}\,day^{-1}}$), mirroring the spatial and temporal patterns of the 1970–2016 precipitation anomaly (see Fig. S.1 and Fig. 8). In particular, a "wet" summer such as occurred in 1997 shows the smallest reduction in isoprene emissions over the Euro-Mediterranean region (-0.16 $\mathrm{mg\,m^{-2}\,day^{-1}}$, -5.09%), while a "dry" summer such as in 2012 displays the largest absolute reduction (-0.48 $\mathrm{mg\,m^{-2}\,day^{-1}}$, -8.11%). Among all simulated summers, the summer 2000 has the largest percentage reduction (-0.47 $\mathrm{mg\,m^{-2}\,day^{-1}}$, -8.41%) since a large dry anomaly occurred in the Balkans, a region characterized by elevated isoprene emissions. The largest decreases in isoprene emissions occur in the summers 2003 and 2012 (Table 4), when the observation-based summer temperatures are nearly 4–5 standard deviations above (warmer than) the 1970–1990 climatology (Fig. S.6). These results suggest that isoprene emissions would be strongly reduced when heat wave and drought co-occur.

Decreases in isoprene emissions drive decreases in formaldehyde concentrations that range between -5 and -20% (Fig. S.7). As shown in Sect.3.1.3, the RegCM4chem-CLM4.5-MEGAN2.1 model underestimates HCHO columns in its standard configuration (Fig. 5). The activation of a soil moisture activity factor in MEGAN (i.e., $\gamma_{SM,2018}$ or $\gamma_{SM,2006}$) decreases isoprene emissions due to the effect of water stress and, as a consequence, decrease HCHO concentrations and does not improve the comparison with OMI HCHO observations. On the contrary, the models used by Wang et al. (2021a), Klovenski et al. (2022) and Wang et al. (2022) overestimate HCHO columns in their standard configurations, hence the indirect effect of the MEGAN soil moisture activity factor on HCHO concentrations improves the comparison between modelled and observed formaldehyde.

## 3.3  Link between anomalies in soil moisture and plant productivity and changes in isoprene emissions

When the MEGAN soil moisture activity factor $\gamma_{SM,2018}$ is activated, dry anomalies limit isoprene emissions. Figure 9 show the spatial distribution of summer anomalies in the total volumetric soil moisture used by vegetation and gross primary productivity (GPP, hereafter also referred to as plant productivity) as simulated by the RegCM4-CLM4.5 1 model for the simulated summers. Spatially, the anomalies in plant productivity overlap and correspond in signs with the anomalies in soil moisture that, in turn, follow the anomalies in precipitation (Fig. 9 and Fig. S.1). Where dry anomalies occur, isoprene emissions decrease when the MEGAN soil moisture activity factor is activated (Fig. 9 vs. Fig. 8).

Focusing over Southern Europe (black box on Fig. 9), Figure 10 shows time series of monthly anomalies in total soil moisture used by vegetation and photosynthesis, and absolute monthly changes in isoprene emissions. All values are averaged over the area: [10°E–30°W; 35°–48°N]. Negative anomalies in soil moisture and plant productivity are in phase with the decrease in isoprene emissions. The summers 2000, 2003, 2012 and 2015 display the most intense dry anomalies in soil moisture, leading to negative anomalies in plant productivity and, subsequently, to important decreases in isoprene emissions (-2 $\mathrm{mg\,m^{-2}\,day^{-1}}$). Among these summers, summer 2012 shows the largest decrease in isoprene emissions, most likely due to the combined effect of the driest anomaly in soil moisture between years 2011 and 2012, followed by a short wet anomaly during spring and an intense dry anomaly during summer. Probably, a lagged effect due to a dry autumn and winter also plays an important role

during summer 2010, when isoprene emissions decrease substantially although both soil moisture and plant productivity show positive summer anomalies. Using meteorological observations and simulations, Vautard et al. (2007) found that hot summers over Europe are correlated with winter-early spring precipitation deficit over Southern Europe.

## 3.4 Effects of changes in isoprene emissions on near surface ozone levels

Under the effect of water stress, the RegCM4chem-CLM4.5-MEGAN2.1 model simulates a reduction in isoprene emissions
that influences levels of near surface ozone. Figure 11 shows the spatial distribution of summer percentage changes in near surface ozone mixing ratio when the water stress factor is activated. Decreases in isoprene emissions reduce near surface ozone levels by less than 2% (Fig. 11; absolute reduction $< 2$ ppbv, Fig. S.8). Over Europe, Jiang et al. (2018) obtained a similar decrease in near surface ozone with a 6-month atmosphere-chemistry-vegetation simulation with the Earth System Model CAM-Chem-CLM4.5-MEGAN2.1 at a coarser resolution ($1.9° \times 2.5°$ spatial res.). Similarly, Guion et al. (2022) simulated
a decrease in near surface ozone of -5% over the Po Valley using the chemistry-transport model CHIMERE. Such a small decrease in near surface ozone levels may be due to a dominant VOC-limited regime, where ozone decreases by lowering VOC emissions, reproduced by the RegCM4chem-CLM4.5-MEGAN2.1 model over the whole domain, as shown in Figure S.9 by computing the ratio between formaldehyde and nitrogen dioxide ($NO2$) based on Duncan et al. (2010).

Spatially, the largest reductions in near surface ozone levels are observed over the southern and eastern part of the Mediter-
ranean Basin (Fig. 11). Multiple modelling studies identified a pronounced land-sea gradient in near surface ozone levels, with higher concentrations over the Mediterranean Basin and the Middle East and lower concentrations over continental Europe and northern Africa (Jaidan et al., 2018, with references therein). In particular, in the eastern Mediterranean Basin, ozone production is favored by the transport of ozone precursors, mainly from the European continent, and by stagnant meteorological conditions dominating during summer (Gerasopoulos et al., 2005). This pattern of near surface ozone levels over the
Mediterranean Basin most likely influences the spatial distribution of changes (Fig. 11).

As for isoprene emissions, the "wet" summer 1997 shows the smallest reduction in near surface ozone levels (-0.06 ppbv, -0.14%). Instead, the largest reduction in near surface ozone, about of -0.17 ppbv (-0.42%), is found in summer 2003, when a heatwave hit Europe (Table 4) due to the temperature effect on ozone chemistry. Indeed, summer 2003 shows the largest and widest warm anomaly in surface air temperatures (Fig. S.6).

## 3.5 Comparison of the two soil moisture activity factors and their effects on isoprene and near surface ozone

The choice of the soil moisture activity factor in MEGAN2.1 influences the pattern and the magnitude of changes in isoprene emissions and near surface ozone levels, as we tested for the summers 1994, 2003 and 2010. Focusing on the summer 2003 when Europe was struck by a series of heatwaves, Figure 12 displays the spatial distribution of the old and new MEGAN soil moisture activity factors, $\gamma_{SM,2006}$ and $\gamma_{SM,2018}$ respectively. The old MEGAN soil moisture activity factor $\gamma_{SM,2006}$ shows
low/mid values between 0 and 0.6 over the Euro-Mediterranean region, with values lower than 0.4 over northern Africa and between 0.4 and 0.6 across Europe. In contrast, the new MEGAN soil moisture activity factor $\gamma_{SM,2018}$ varies across the whole scale of values (from 0 to 1), shows some areas where there is no water stress (in white), and localized areas with mid values

(between 0.3 and 0.7). For example, over Italy and the Balkans, which show the largest soil moisture anomalies during summer 2003 (see Fig. 9), $\gamma_{SM,2006}$ has a homogeneous pattern with values between 0.4 and 0.5, while $\gamma_{SM,2006}$ has values between 0.5 and 0.7 over small areas.

The pattern and magnitude of the soil moisture activity factor influence the spatial distribution of changes in isoprene emissions (Fig. 13). Using the old soil moisture activity factor $\gamma_{SM,2006}$, isoprene emissions decrease by more than 25% over the Euro-Mediterranean region, with absolute decrease spanning from -0.5 up to -12 $\mathrm{mg\,m^{-2}\,day^{-1}}$. Once averaged over the Euro-Mediterranean region, decreases in isoprene emissions are around -3 $\mathrm{mg\,m^{-2}\,day^{-1}}$ (-57%, Table 4). Over Australia and China, Emmerson et al. (2019) and Wang et al. (2021b) found similar decrease in isoprene emissions using the Guenther et al. (2006)'s parameterization (i.e., $\gamma_{SM,2006}$). Conversely, when applying the new soil moisture activity factor $\gamma_{SM,2018}$, isoprene emissions decrease by much lower magnitudes, with percentage changes between -5 and -25% and larger reduction in the Middle East ($\Delta_{\%}ISOP < -75\%$). Over most of the Euro-Mediterranean region, decreases in isoprene emissions are smaller than 2.5% ($> -0.25\,\mathrm{mg\,m^{-2}\,day^{-1}}$).

Concerning the effects on near-surface ozone, when applying the old soil moisture activity factor, near-surface ozone levels decrease homogeneously by $\sim 10\%$ (between -1.2 and 7.5 $\mathrm{ppbv}$) over the Euro-Mediterranean region, with reductions between -2.5 and -5% over Europe and between -5 and -10% over the Mediterranean Basin (Fig. 14). Once averaged over the Euro-Mediterranean region, decreases in near surface ozone mixing ratio are around -3 $\mathrm{ppbv}$ (-4%, Table 4). Similarly, using the Guenther et al. (2006)'s parameterization (i.e., $\gamma_{SM,2006}$) over China, Wang et al. (2021a) found that decreases in isoprene emissions reduce near surface ozone up to -8% in drought-hit regions and during dry years. Conversely, when using the new soil moisture activity factor, near surface ozone levels are not reduced by more than -1.5%.

## 4   Discussion and conclusions

The water stress effects on isoprene emissions and near surface ozone mixing ratios were investigated for selected dry and wet summers in the Euro-Mediterranean region, using a regional vegetation–chemistry–climate model. This model includes the BVOC emission model MEGAN which provides two parameterizations limiting isoprene emissions under water stress. By performing sensitivity simulations, we assessed the decrease in isoprene emissions due to water stress and its impact on near surface ozone mixing ratios. Moreover, we compared the two parameterizations available in MEGAN: the Guenther et al. (2006)'s and the Jiang et al. (2018)'s parameterizations.

Our results show that, over the Euro-Mediterranean region, water stress reduces isoprene emissions on average by 6%, with larger decreases of up to 40–60% over sensitive areas (e.g., the Balkans) and during very dry summers (e.g., 2003, 2015). Sustained decreases in isoprene emissions not only co-occur with negative anomalies in precipitation, soil moisture and plant productivity, but are also influenced by the lagged effect of prolonged or repeated dry anomalies, as observed for the summers 2010 and 2012. This result confirms that it is critical to correctly initialize soil moisture for atmospheric-chemistry-vegetation numerical experiments. Regarding the indirect impact of water stress on near surface ozone, our results suggest that over the Euro-Mediterranean region near surface ozone levels have a limited sensitivity to decreases in isoprene emissions, with

reductions of near surface ozone by a few percents, most likely due to a dominant VOC-limited regime over the region, in agreement with (Jiang et al., 2018). When comparing the two MEGAN parameterizations of water stress impact on isoprene emissions, we found substantial differences in the reduction of both isoprene emissions and near surface ozone mixing ratios. Compared to the Guenther et al. (2006)'s parameterization, the Jiang et al. (2018)'s parameterization leads to more localized and 25–50% smaller decreases in isoprene emissions, and 3–8% smaller reduction in near surface ozone mixing ratios.

To summarize, our results (1) confirm the importance of soil moisture initialization when performing atmospheric-chemistry-vegetation numerical experiments, (2) show that the choice of the MEGAN parameterization to account for water stress impacts on isoprene emissions produces different reductions in ozone levels, and (3) suggest that the indirect effect of water stress on ozone levels via changes in isoprene emissions is limited over the Euro-Mediterranean region. Our results are partly influenced by the prevailing cold temperature and wet precipitation biases identified for the RegCM4-CLM4.5 model. Warm atmospheric conditions favor both isoprene emissions and the production of near surface ozone, hence a model characterized by a cold-wet bias most likely has a negative bias in reproducing isoprene emissions and near surface ozone levels. Nevertheless, our results compare well with similar studies (Jiang et al., 2018; Emmerson et al., 2019; Wang et al., 2021b; Klovenski et al., 2022). While we observed an opposite sensitivity of isoprene emissions to the two parameterizations compared to the study by Guion et al. (2022), who obtained larger decrease in isoprene emissions (-39%) when using an adapted formulation of Jiang et al. (2018) and a smaller decrease (-12%) using the Guenther et al. (2006)'s parameterizations. This result confirms the conclusions of Klovenski et al. (2022) who highlighted the need to tune the MEGAN soil moisture activity factor, and its parameters, based on the modelling set-up, which influences the soil moisture activity factor. In a future study, we aim to explore the ozone climate penalty over the Euro-Mediterranean region under both present-day and future climates and to assess the impact on ozone concentration of both the direct effect of high temperatures and the indirect effect of water stress on isoprene emissions.

To isolate the water stress impact on isoprene emissions and near surface ozone, we chose to not account for aerosols (natural and anthropogenic) and for the chemistry feedback on climate. A fully coupled vegetation–chemistry–climate model may produce different results, with aerosols (time-varying or constant) influencing surface solar radiation, surface air temperature, cloud cover, evapotranspiration over land and lastly relative humidity (Boé et al., 2020), with consequences on the thermal and water stresses experienced by plants, which influence vegetation processes and isoprene emissions. Moreover, water stress could potentially impact other processes involved in atmospheric chemistry such as the formation of secondary organic aerosols (SOAs) that plays an important role in the Euro-Mediterranean region (Freney et al., 2018, and references therein). Hence, future studies should apply fully coupled vegetation–chemistry–climate simulations to assess the overall effect of water stress on isoprene emissions and ozone levels when accounting for the chemistry feedback on climate, and as well the feedback of ozone levels on plant physiology and productivity (e.g., Yue et al., 2017). Moreover, water stress could potentially impact other processes involved in atmospheric chemistry such as the formation of secondary organic aerosols (SOAs) that plays an important role in the Euro-Mediterranean region (Freney et al., 2018, and references therein). These simulations could use regional inventories of emission factors that, compared to the PFT-based method, can reproduce regional variations in the emission potentials of vegetation, as shown over France by Solmon et al. (2004). In addition, to assess the effect of water stress on ozone levels, it would be important to directly link the modelling of soil moisture and ozone deposition velocity

to compare the effect of water stress on isoprene emissions with the effect of water stress on the ozone dry deposition. Due to high temperatures and precipitation deficit, plants can undergo water stress, thus reducing ozone dry deposition (Vautard et al., 2005). This process could counteract the decrease in isoprene emissions, which drives the decrease in near surface ozone levels. Via sensitivity simulations over south-western Europe, Guion et al. (2022) evaluated the combined effect on ozone levels of decrease in isoprene emissions and ozone dry deposition. Using a chemistry-transport model, Guion et al. (2022) obtained a slight increase in ozone levels, suggesting that the decrease in dry deposition is the dominant process. However, model performance in reproducing ozone dry deposition also depends on the meteorological forcing, which is simulated offline in a chemistry-transport model. Moreover, the modelling of ozone chemistry strongly depends on the spatial resolution that influences the model ability in adequately distinguish chemical regimes (i.e., VOC- or NOx- limited) that, in turn, depend on the emission pattern of natural and anthropogenic sources (Massad et al., 2019). Hence, future studies should explore water stress effect on isoprene and ozone in urban contexts, where more VOC-limited conditions dominate.

Both the Guenther et al. (2006)'s and the Jiang et al. (2018)'s parameterizations have two limits. Firstly, both parameterizations rely on the assumption that water stress limits isoprene (and in general BVOCs) emissions, an hypothesis that is still under debate ((Strada et al., 2023) and references therein). Recently, Wang et al. (2021a) tested a parameterization which increases isoprene emissions under mild water stress via changes in leaf temperature, and when applied over China, this parameterization increases isoprene emissions up to 30%. Secondly, in addition to the assumption that water stress only reduces isoprene emissions, MEGAN only parameterizes the impact of water stress on isoprene emissions, not on monoterpenes, i.e. compounds in the BVOC family largely emitted in the Mediterranean region that contribute to the production of secondary organic aerosols and that seem to behave differently from isoprene under water stress (Feng et al., 2019). Hence, the need to improve our understanding of the role of water availability in the BVOC production and emission and to realistically represent plant processes in a changing climate calls for more observational studies on the soil water dependence of BVOC emissions. These can take advantage and combine new data-sets, from the next-generation machine learning isoprene retrievals (Wells et al., 2022), longer time-series of soil moisture content from multiple satellite-based data-sets (e.g., the NASA Soil Moisture Active Passive instrument, SMAP, and the European Space Agency Soil Moisture and Ocean Salinity mission, SMOS), and new proxies of plant productivity that do not simply rely on greenness (e.g., the sun-induced chlorophyll fluorescence, SIF; Pagán et al., 2019; Walther et al., 2019).

*Author contributions.* S. S., A. P., G. G., E. C., F. S. and F. G. designed the numerical experiments. S. S. and G. G. modified the model code with support from F. S., A. P., A. G. and X. J. S. S. performed simulations and analysis. S. S. prepared the manuscript with contributions from all co-authors.

*Code and data availability.* Source code available at: https://github.com/ICTP/RegCM/releases/tag/IDIOM, Licence: GPL3. This paper makes use of different datasets, here we acknowledge all the data providers and specify the data availability: the EU-FP6 project UERRA, the

Copernicus Climate Change Service, and the data contributors in the ECA&D project for making the E-OBSv20e data-set available at https://www.ecad.eu/download/ensembles/download.php; the EUMETSAT within the CM-SAF project for providing the CLAASv1 data-set, available at https://wui.cmsaf.eu/safira/action/viewDoiDetails?acronym=CLAAS_V001; the FLUXCOM initiave for providing latent heat fluxes (Remote-Sensed product) available at https://www.fluxcom.org/; the ESA-CCI SM project for providing surface soil moisture (ESA-CCI SSM v04.4, COMBINED product) available at https://esa-soilmoisture-cci.org/node/236; the EU FP7 project Quality Assurance for Essential Climate Variables for providing OMI data of formaldehyde column concentrations (FP7-SPACE-2103-1, Project No 607405) available at http://www.qa4ecv.eu; and the European Environment Agency (EEA) for providing near surface ozone measurements, available at https://www.eea.europa.eu/data-and-maps/data/aqereporting-9. In-situ isoprene and ozone measurements from the field site La Verdière (France) are available at: https://vocsnetdata.ceh.ac.uk/page/login.aspx. In-situ isoprene measurements from the field site Ineia (Cyprus) are available at: https://zenodo.org/record/8267184.

*Competing interests.* A. Pozzer is an editor of the journal.

*Acknowledgements.* This project has received funding from the European Union's Horizon 2020 research and innovation programme under grant agreement No.: 791413 (project IDIOM2). A. Guenther and X. Jiang were supported by the US National Science Foundation award number AGS-1643042. Isoprene and ozone measurements from the field site La Verdière (France) were made available with support from the ESCOMPTE program. E. Bourtsoukidis and J. Williams acknowledge the support from the European Union's Horizon 2020 research and innovation programme under grant agreement No.: 856612 (project EMME-CARE).

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

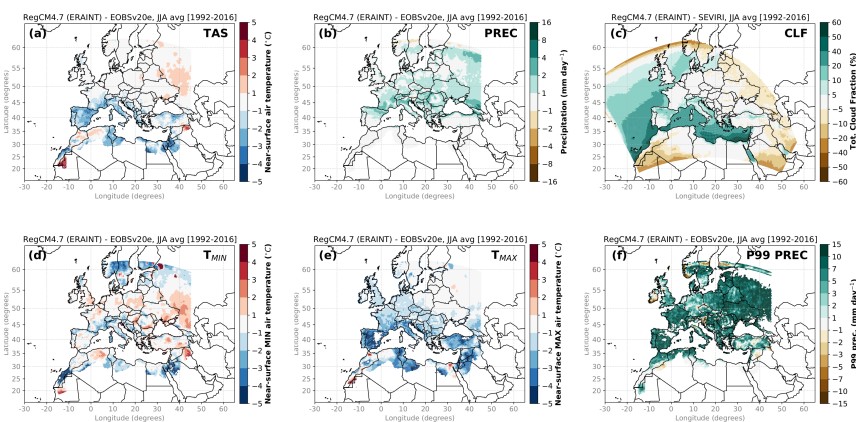

**Figure 1.** Spatial distribution of summer (June–July–August, JJA) biases over the Med-CORDEX domain and over the period 1992–2016 in: (a) 2-m air temperature (units: °C), (b) precipitation rate ($\mathrm{mm\,day^{-1}}$), (c) total cloud fraction (unitless), (d)-(e) daily maximum and minimum 2-m air temperature (°C) and (f) the $99^{th}$ percentile of precipitation rate (P99 prec.; $\mathrm{mm\,day^{-1}}$). Biases correspond to differences between summer averaged fields from the RegCM4-CLM4.5 model and: (a, b, d–f) the E-OBSv20e data-set for temperature and precipitations and (c) the CLAAS data-set for total cloud fraction. For comparison, model output was remapped onto the observation grid.

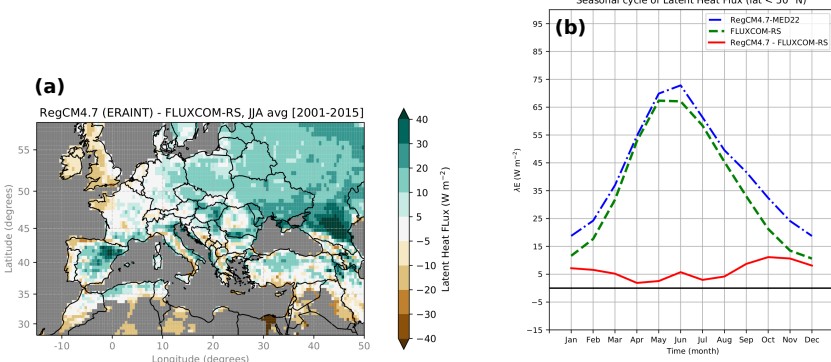

**Figure 2.** Comparison of latent heat fluxes (units: $\mathrm{W\,m^{-2}}$) between the RegCM4-CLM4.5 model and the FLUXCOM data-set (Remote-Sensed product) over 2001–2015: (a) spatial distribution of summer bias; (b) annual cycles southern than 50°N from the RegCM4-CLM4.5 model (blue line), the FLUXCOM data-set (green line), and their difference (red line) as computed over the illustrated part of the model domain, hereafter referred to as the Euro-Mediterranean region (coordinates: [15°E–50°W; 28°–58°N]). Biases correspond to differences between summer averaged fields from the RegCM4-CLM4.5 model and the FLUXCOM data-set. For comparison, model output was remapped onto the FLUXCOM grid.

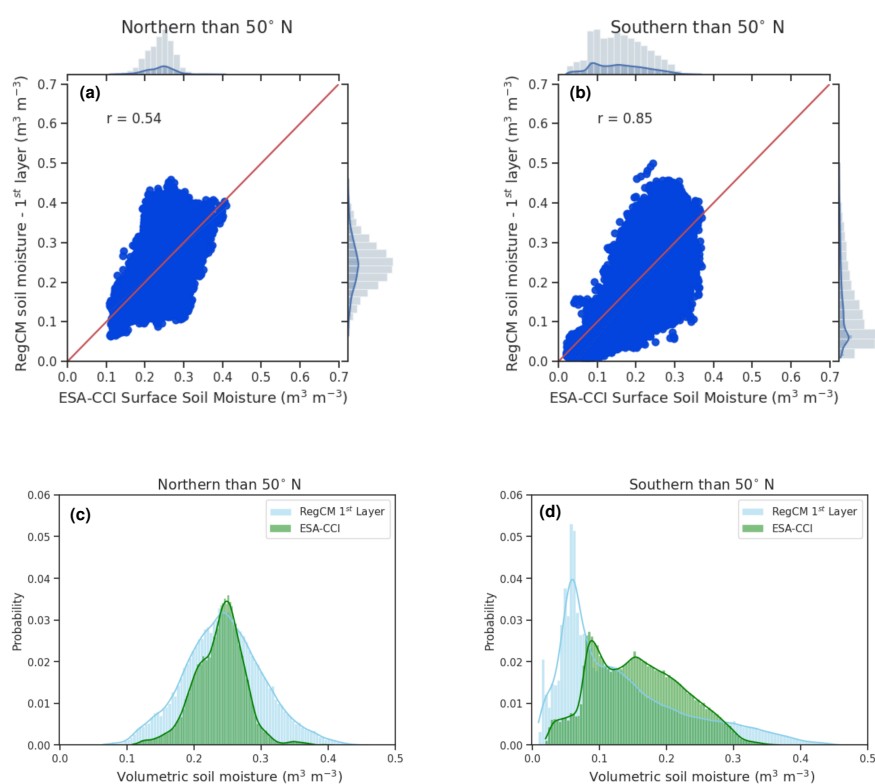

**Figure 3.** Comparison of volumetric soil moisture (unit: $\mathrm{m^3\ m^{-3}}$) between the the $1^{st}$ soil layer in the RegCM4-CLM4.5 model and the ESA-CCIv4.04 data-set over the period 2005–2015: (a) and (b) cloud plots and Pearson's $r$ correlations between the RegCM4-CLM4.5 model (Y-axis) and the ESA-CCIv4.04 data-set (X-axis); (c) and (d) probability distributions of soil moisture from the RegCM4-CLM4.5 model (in blue) and the ESA-CCIv4.04 data-set (in green). Plots (a) and (c) refer to the area northern than 50°N, and plots (b) and (d) to the area southern than 50°N in Fig. 2.a. For comparison, model output was remapped onto the ESA-CCI grid.

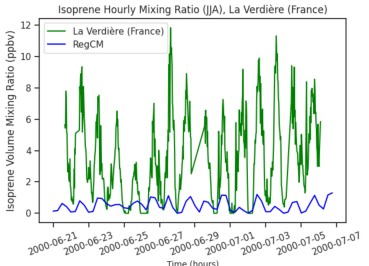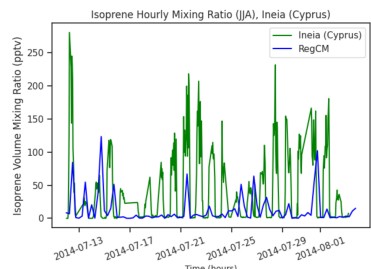

**Figure 4.** Comparison of the time-series of isoprene concentrations collected at La Verdière (Latitude: 43.63° N; Longitude: 5.93° E; France, summer 2000; units: ppbv) and at Ineia (Latitude: 34.96° N; Longitude: 32.39° E; Cyprus, summer 2014; units: pptv). The green solid line shows observations, while the blue solid line shows the model output extracted over the nearest grid-cell to the observation spot from the GAMMA-SMoff simulation performed with the RegCM4chem-CLM4.5-MEGAN2.1 model.

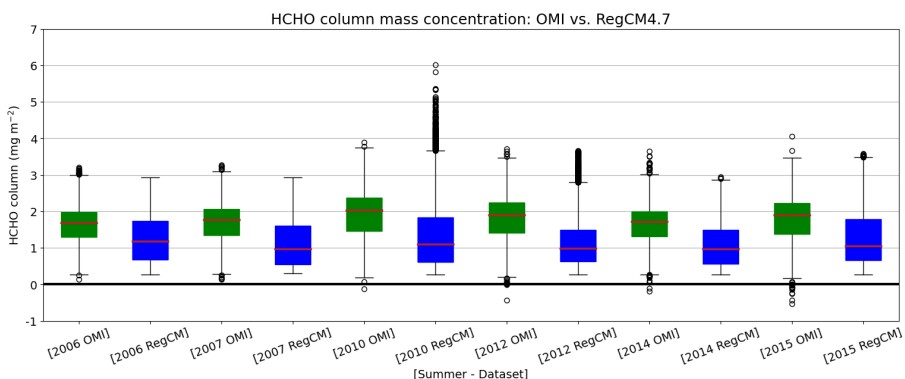

**Figure 5.** Box-and-whisker plots showing the distribution of summer-averaged mass column concentrations of formaldehyde (HCHO) (units: $\mathrm{mg\,m^{-2}}$) as observed by the Ozone Monitoring Instrument (OMI, version QA4ECV; green boxes) and as reproduced by the RegCM4chem-CLM4.5-MEGAN2.1 (blue boxes) over land-only grid-cells in the Euro-Mediterranean region (see Fig. 2.a) and across simulated summers over the period 2005–2016. Boxes display the interquartile range ($Q_{25}$, $Q_{75}$), with the orange line showing the median value; whiskers cover from $Q_{25} - 1.5 \times (Q_{75} - Q_{25})$ to $Q_{75} - 1.5 \times (Q_{75} - Q_{25})$; empty black circles represent outliers. Negative values in the OMI data-set reflect the high noise in the data detected over scenes with low HCHO columns. For comparison, model output has been remapped onto the OMI grid.

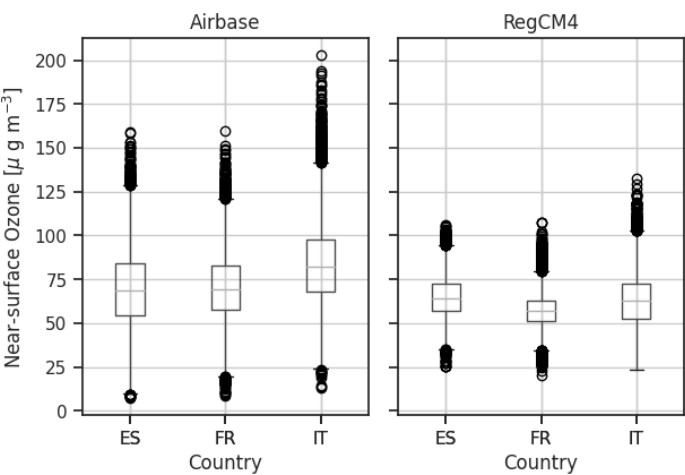

**Figure 6.** Box-and-whisker plots show the distribution of summer daily mass concentrations of near surface ozone levels (O3) (units: $\mu$ g m$^{-3}$) as observed by the Airbase monitoring network (on the left) and as reproduced by the RegCM4chem-CLM4.5-MEGAN2.1 (on the right) over the summer 2015 over Spain (ES), France (FR) and Italy (IT). Boxes display the interquartile range ($Q_{25}$, $Q_{75}$), with the orange line showing the median value; whiskers cover from $Q_{25} - 1.5 \times (Q_{75} - Q_{25})$ to $Q_{75} - 1.5 \times (Q_{75} - Q_{25})$; empty black circles represent outliers. For comparison, model output have been taken at the closest grid-cell to each Airbase station.

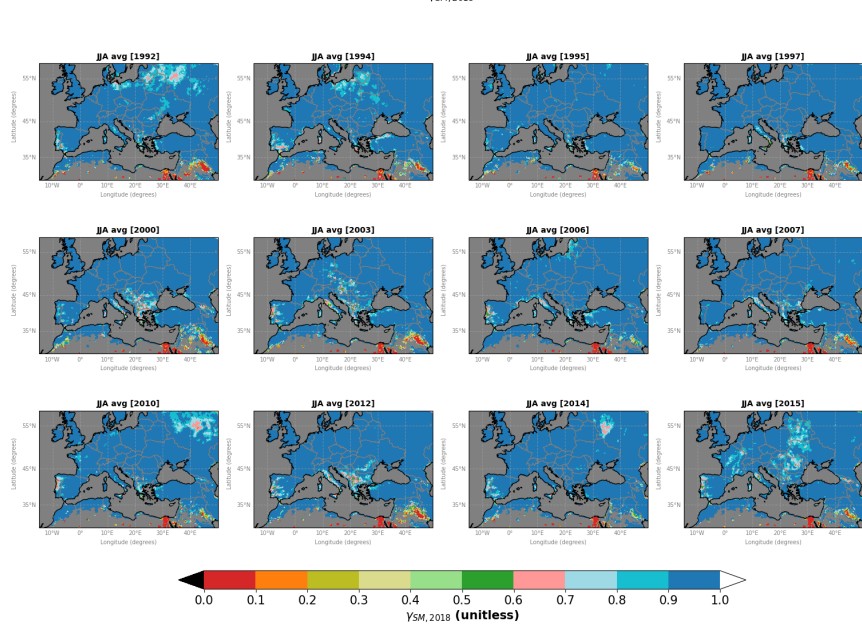

**Figure 7.** Spatial distribution of summer-averaged soil moisture activity factor $\gamma_{SM,2018}$ (unitless) as simulated by the RegCM4chem-CLM4.5-MEGAN2.1 model over the Euro-Mediterranean region and across selected summers over 1992–2016. The $\gamma_{SM,2018}$ factor ranges between 0 (severe water stress) to 1 (no water stress) and was derived dividing isoprene emissions from the GAMMA-SM2018on simulation by emissions from the reference simulation GAMMA-SMoff. Grey areas correspond to grid-cells where isoprene emissions are not defined.

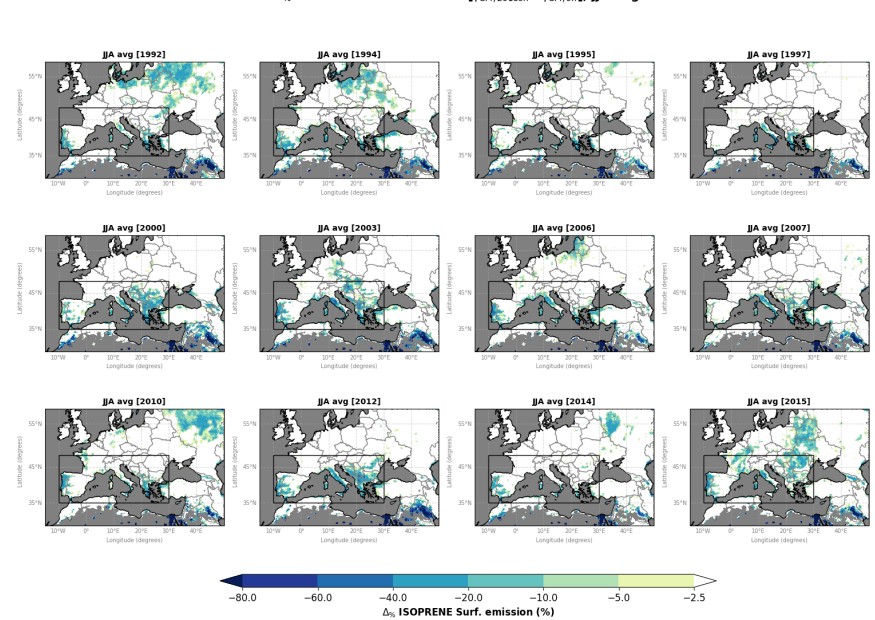

**Figure 8.** Spatial distribution of summer-averaged percentage changes in isoprene emissions (units: %) as simulated by the RegCM4chem-CLM4.5-MEGAN2.1 model over the Euro-Mediterranean region and across selected summers over 1992–2016. To compute percentage changes, the difference between JJA averages from the GAMMA-SM2018on ($\gamma_{SM,2018}$) and the GAMMA-SMoff simulations was divided by the reference simulation GAMMA-SMoff. Black boxes highlight the area of Southern Europe selected for further analysis.

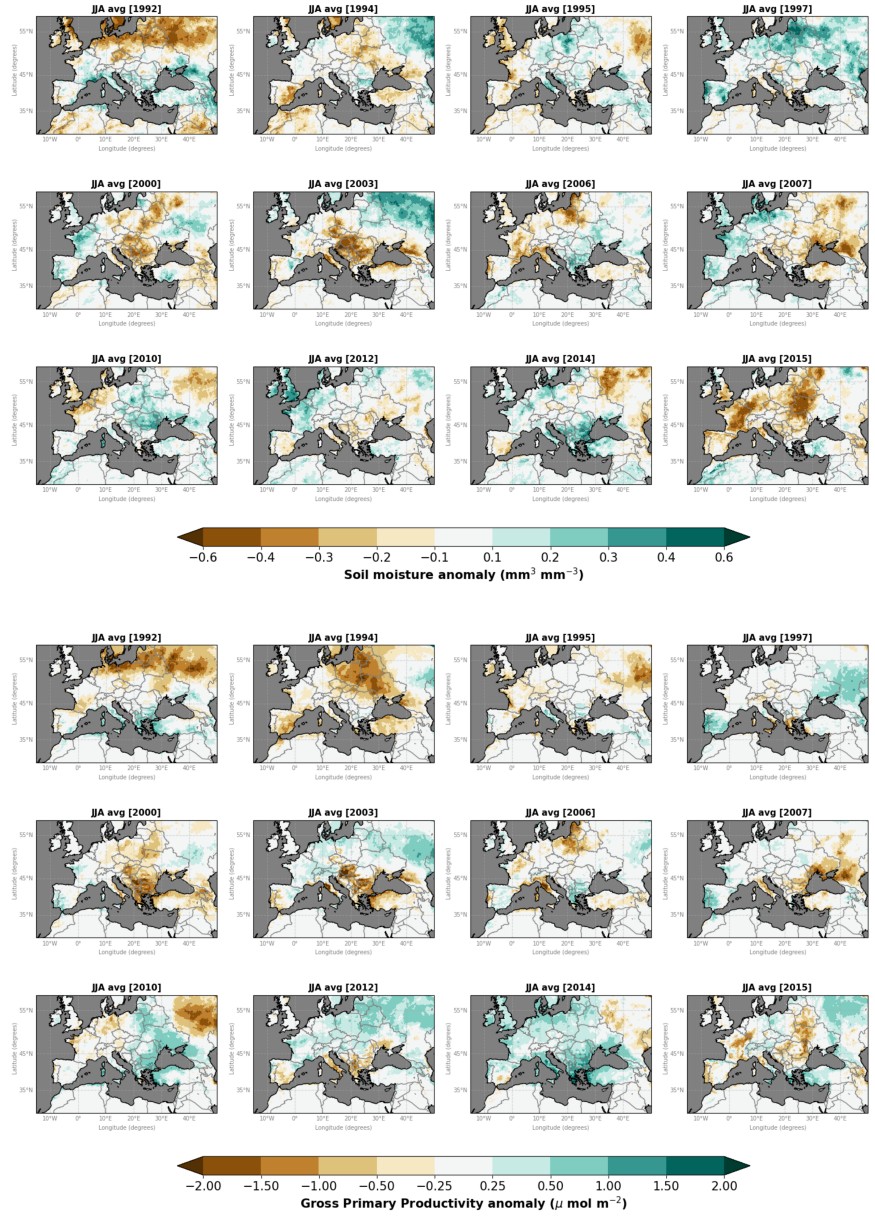

**Figure 9.** Spatial distribution of summer anomalies in soil moisture used by vegetation (units: $\mathrm{mm^3\ mm^{-3}}$ and gross primary productivity (GPP, units: $\mu\mathrm{mol\ m^{-2}}$) as simulated by the RegCM4-CLM4.5 model over the Euro-Mediterranean region and across selected summers over 1992–2016.

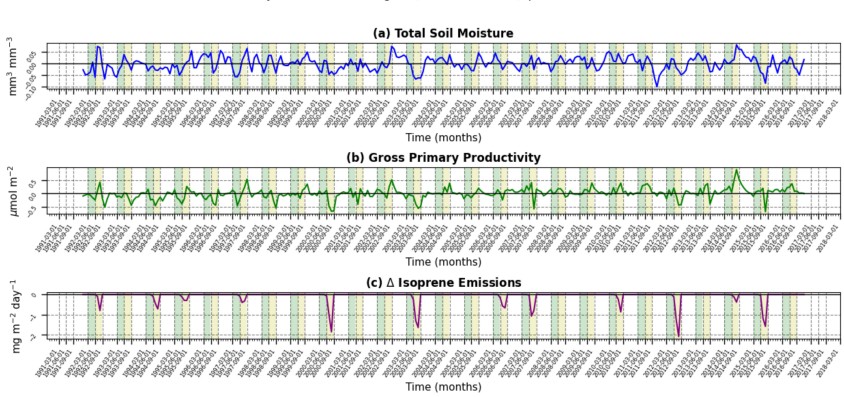

**Figure 10.** Time series of monthly anomalies in: (a) total soil moisture used by vegetation (units: $\mathrm{mm}^3\ \mathrm{mm}^{-3}$; blue line), (b) gross primary productivity (GPP, units: $\mu\mathrm{mol}\,\mathrm{m}^{-2}$, green line), and (c) absolute change in monthly means of isoprene emissions (units: $\mathrm{mg}\,\mathrm{m}^{-2}\,\mathrm{day}^{-1}$; purple line). Anomalies and means were computed over the central part of the domain (black box on Figure 8; coordinates: [10°E–30°W; 35°–48°N]). Absolute changes in isoprene emissions correspond to the difference between monthly averages from the GAMMA-SM2018on and the reference simulation, GAMMA-SMoff, across selected summers over 1992–2016. Green (yellow) stripes highlight spring, March–April–May, (summer, June–July–August) season.

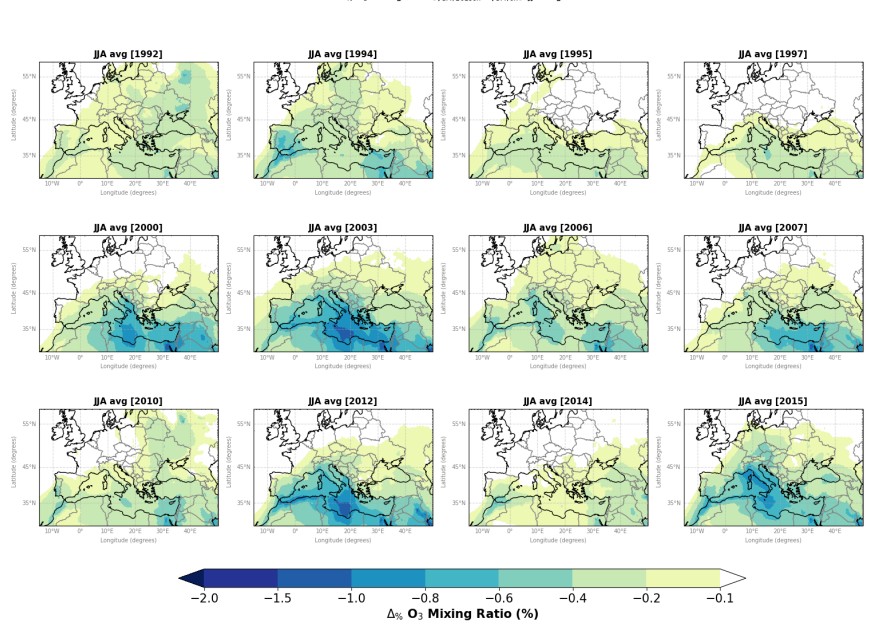

**Figure 11.** Spatial distribution of summer-averaged percentage changes in ozone ($O_3$) mixing ratio at 1000 hPa (units: %) as simulated by the RegCM4chem-CLM4.5-MEGAN2.1 model over the Euro-Mediterranean region and across selected summers over 1992–2016. To compute percentage changes, the difference between summer averages from the GAMMA-SM2018on and the GAMMA-SMoff simulations was divided by the reference simulation, GAMMA-SMoff.

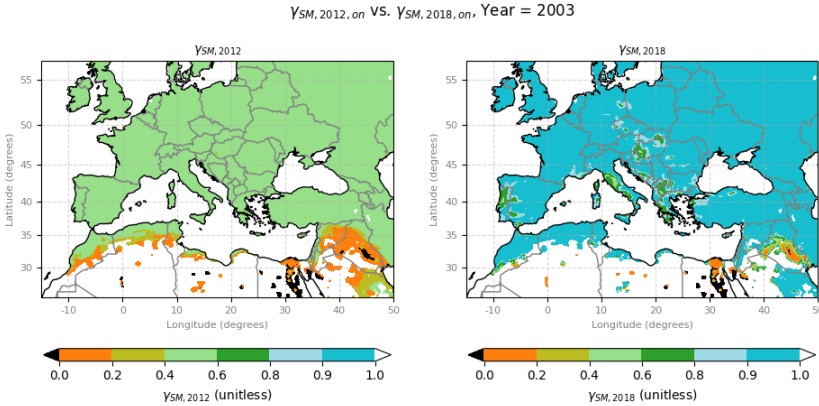

**Figure 12.** Spatial distribution of summer-averaged soil moisture activity factors $\gamma_{SM,2006}$ (left) and $\gamma_{SM,2018}$ (right) (unitless) over the Euro-Mediterranean region and for the summer 2003.

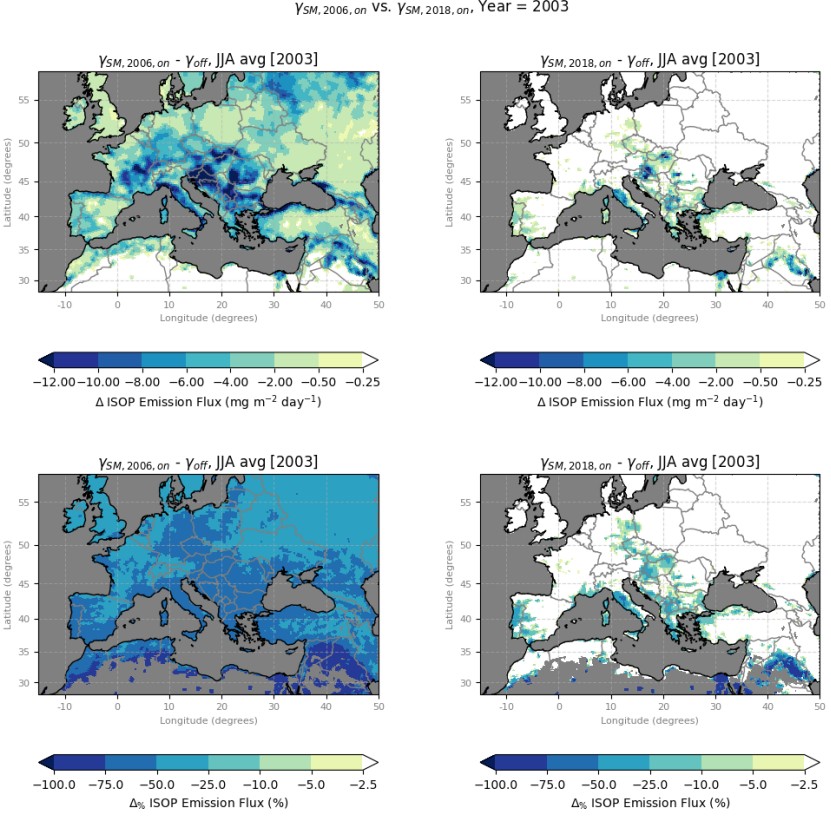

**Figure 13.** Spatial distribution of summer-averaged absolute (units: $\mathrm{mg\,m^{-2}\,day^{-1}}$) and percentage (%) changes in isoprene emissions computed over the Euro-Mediterranean region and for the summer 2003 using the $\gamma_{SM,2006}$ and the $\gamma_{SM,2018}$ soil moisture activity factors. Changes correspond to differences between summer averages of model output from the GAMMA-SMon and the GAMMA-SMoff simulations using the old ($\gamma_{SM,2006}$, GAMMA-SM2006on) or the new ($\gamma_{SM,2018}$, GAMMA-SM2018on) soil moisture activity factor.

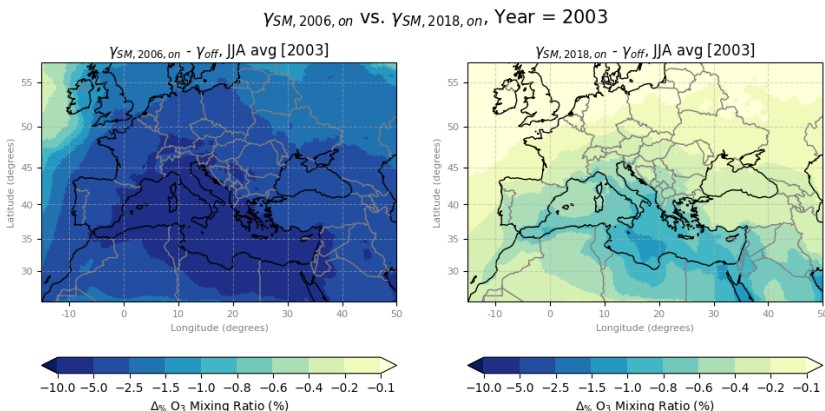

**Figure 14.** As Figure 13, spatial distribution of summer-averaged percentage changes (units: %) in ozone mixing ratio at 1000 hPa for the summer 2003.

**Table 1.** Non-methane volatile organic compounds speciation based on fractions of total emitted carbon as obtained from Table 1 in Szopa et al. (2005) or from other sources, when specified.

| Chemical species | Long name | Fraction of total emitted carbon | Notes |
|---|---|---|---|
| C2H6 | Ethane | 0.0163 | |
| CH3OH | Methanol | 0.0050 | |
| ACET | Ketones | 0.0415 | methyl-ethyl-ketone + methyl-isobutyl-ketone = 0.0303 + 0.0112 |
| ALD2 | Acetaldehyde | 0.0009 | |
| AONE | Acetone | 0.0165 | |
| ETH | Ethylene (C2H4) | 0.0309 | |
| HCHO | Formaldehyde | 0.0047 | |
| OLET | Terminal olefin carbons | 0.0154 | but-1-ene + pent-1-ene + 2methylbut-1-ene + 3methylbut-1-ene = 0.007 + 0.0038 + 0.0021 + 0.0025 |
| OLEI | Internal olefin carbons | 0.0384 | propene + but-2-ene + pent-2-ene + 2-methylbut-2-ene = 0.0111 + 0.0129 + 0.0097 + 0.0047 |
| PAR | Paraffin carbons | 0.3324 | See Table 2 |
| RCOOH | Higher organic acids | 0.031 | from Pozzer et al. (2007) |
| TOL | Toluene | 0.1043 | |
| XYL | Xylene | 0.0865 | o-xylene + m-xylene + p-xylene = 0.0241 + 0.0312 + 0.0312 |

**Table 2.** List of paraffin carbons and their related fraction of alkane compounds (Tab. 4 in Zaveri and Peters, 1999), and fraction of total emitted carbon (Tab. 1 in Szopa et al., 2005).

| Chemical species | Fraction of alkane compounds | Fraction of total emitted carbon | Notes |
|---|---|---|---|
| Propane | 0.0975 | 0.004 | |
| n-butane | 0.2863 | 0.0764 | |
| Iso-butane | 0.0777 | 0.0415 | |
| n-pentane | 0.0473 | 0.0287 | |
| Iso-pentane | 0.0897 | 0.0444 | |
| 2,2-dimethylpropane | 0.0171 | Not available | |
| n-hexane | 0.0687 | 0.0197 | |
| 2-methylpentane | 0.039 | 0.0223 | |
| 3-methylpentane | 0.0231 | 0.0157 | |
| 2,2-dimethylbutane | 0.0078 | 0.0022 | |
| 2,3-dimethylbutane | 0.0154 | 0.0071 | |
| n-heptane | 0.069 | 0.0104 | |
| 2-methylhexane | 0.0044 | 0.0087 | |
| 3-methylhexane | 0.0128 | 0.0075 | |
| 2,4-dimethylpentane | 0.0101 | Not available | |
| n-octane | 0.026 | 0.0076 | |
| 2,2,4-trimethylpentane | 0.0426 | Not available | |
| 2,3,3-trimethylpentane | 0.0239 | Not available | |
| 4-methylheptane | 0.01 | 0.0088 | 2.64% in Szopa et al. (2005) equally splitted between 2-, 3- and 4-methylheptane |
| 3-methylheptane | 0.0084 | 0.0088 | 2.64% in Szopa et al. (2005) equally splitted between 2-, 3- and 4-methylheptane |
| n-dodecane | 0.0226 | 0.0186 | |
| bf Total | 0.9994 | 0.3324 | |

**Table 3.** Summary of performed simulations.

| Name | Period/Summers | Chemistry | $\gamma_{\mathrm{SM}}$ |
|---|---|---|---|
| ATM | 1992–2016 (Spin-up: 1987–1991) | — | — |
| GAMMA-SMoff | Summers: | ✓ | — |
| GAMMA-SM2018on | 1992, 1994, 1995, 1997, 2000, 2003, 2006, 2007, 2010, 2012, 2014, 2015 | ✓ | $\gamma_{SM,2018}$ |
| GAMMA-SM2006on | Summers: 1994, 2003, 2010 | ✓ | $\gamma_{SM,2006}$ |

**Table 4.** Absolute and percentage changes in summer isoprene emissions and near surface ozone mixing ratios averaged over the Euro-Mediterranean region (see Fig. 2.a) and computed across the simulated summers. Changes correspond to the difference between summer means from the GAMMA-SM2018on (or GAMMA-SM2006on) and the GAMMA-SMoff simulations. Percentage changes were obtained by dividing absolute changes by the reference simulation, GAMMA-SMoff.

| Summer | Δ Isoprene emissions $(\mathrm{mg\,m^{-2}\,day^{-1}}$ [%]) | | Δ Near surface ozone mixing ratio (ppbv [%]) | |
| --- | --- | --- | --- | --- |
| | $\gamma_{SM,2018}$ | $\gamma_{SM,2006}$ | $\gamma_{SM,2018}$ | $\gamma_{SM,2006}$ |
| 1992 | -0.292 [-7.829] | — | -0.080 [-0.204] | — |
| 1994 | -0.311 [-7.453] | -2.671 [-57.205] | -0.105 [-0.266] | -1.464 [-3.707] |
| 1995 | -0.169 [-5.158] | — | -0.065 [-0.161] | — |
| 1997 | -0.161 [-5.085] | — | -0.058 [-0.144] | — |
| 2000 | -0.466 [-8.412] | — | -0.129 [-0.326] | — |
| 2003 | -0.437 [-8.185] | -2.799 [-56.848] | -0.173 [-0.424] | -1.681 [-4.206] |
| 2006 | -0.256 [-5.584] | — | -0.106 [-0.266] | — |
| 2007 | -0.319 [-6.025] | — | -0.105 [-0.266] | — |
| 2010 | -0.368 [-6.047] | -3.258 [-56.185] | -0.088 [-0.225] | -1.639 [-4.181] |
| 2012 | -0.476 [-8.106] | — | -0.141 [-0.364] | — |
| 2014 | -0.202 [-4.647] | — | -0.061 [-0.155] | — |
| 2015 | -0.470 [-6.892] | — | -0.136 [-0.354] | — |

# Assessment of isoprene and near surface ozone sensitivities to water stress over the Euro-Mediterranean region

Strada, S.[1], Pozzer, A., Giorgi, F., Giuliani, G., Coppola, E., Solmon, F., Jiang, X., Guenther, A.

## Supplementary Material

[1] *Corresponding author*: sstrada@ictp.it

**Table S.1.** Plant Functional Type classes (PFTs) in the Community Land surface model version 4.5 (CLM4.5, Oleson et al., 2013).

| PFT name |
| --- |
| 1. Bare soil |
| 2. Needleleaf Evergreen Tree - Temperate |
| 3. Needleleaf Evergreen Tree - Boreal |
| 4. Needleleaf Deciduous Tree - Boreal |
| 5. Broadleaf Evergreen Tree - Tropical |
| 6. Broadleaf Evergreen Tree - Temperate |
| 7. Broadleaf Deciduous Tree - Tropical |
| 8. Broadleaf Deciduous Tree - Temperate |
| 9. Broadleaf Deciduous Tree - Boreal |
| 10. Broadleaf Deciduous Shrub - Temperate |
| 11. Broadleaf Evergreen Shrub - Temperate |
| 12. Broadleaf Deciduous Shrub - Boreal |
| 13. C3 artic grass |
| 14. C3 grass |
| 15. C4 grass |
| 16. Crop 1 |
| 17. Crop 2 |

**Table S.2.** Summary of observation-based data-sets used in the present study.

| Dataset (version) | Variable | Units | Spatial res. | Period | Temporal res. | Reference |
|---|---|---|---|---|---|---|
| E-OBS v20e | Surface air temperature | °C | 0.25° | 1950–2018 | Daily Mean | Cornes et al. (2018) |
| | Precipitation rate | mm day$^{-1}$ | 0.25° | 1950–2018 | Daily Mean | Cornes et al. (2018) |
| CLoud property dAtAset using SEVIRI, version 1 (CLAASv1) | Fractional cloud cover | % | 0.05° | 1991–2015 | Monthly Mean | Stengel et al. (2014) |
| FLUXCOM remote-sensed (RS) product) | Latent heat flux | MJ m$^{-2}$ d$^{-1}$ | 0.50° | 2001–2015 | Monthly Mean | Jung et al. (2019) |
| European Space Agency Climate Change Initiative (ESACCIv4.04) COMBINED product | Volumetric surface soil moisture | m$^3$ m$^{-3}$ | 0.25° | 1978–2015 | Daily Mean | Dorigo et al. (2017) |
| Ozone Monitoring Instrument (OMI-L3 vQA4ECV) | Formaldehyde (HCHO) column concentration | $10^{15}$ molec cm$^{-2}$ | 0.25° | 2005–2015 | Monthly mean | De Smedt et al. (2018) |
| European Air quality Database (AirBase) | Mixing ratio | ppbv | | 200*–200* | Daily mean | |

**Table S.3.** For each soil layer in RegCM4.7, inferior bound and thickness.

| Soil layer number | Soil inferior bound (m) | Soil thickness (m) |
| --- | --- | --- |
| 1 | 0.0175 | 0.0175 |
| 2 | 0.0451 | 0.0276 |
| 3 | 0.0906 | 0.0455 |
| 4 | 0.1655 | 0.0750 |
| 5 | 0.2891 | 0.1236 |
| 6 | 0.4929 | 0.2038 |
| 7 | 0.8289 | 0.3360 |
| 8 | 1.3828 | 0.5539 |
| 9 | 2.2961 | 0.9133 |
| 10 | 3.8019 | 1.5058 |

**JJA Standardized Anomaly, E-OBSv20ens Mean Prec (1970-2016 vs. 1970-1990), Res. 0.25°**

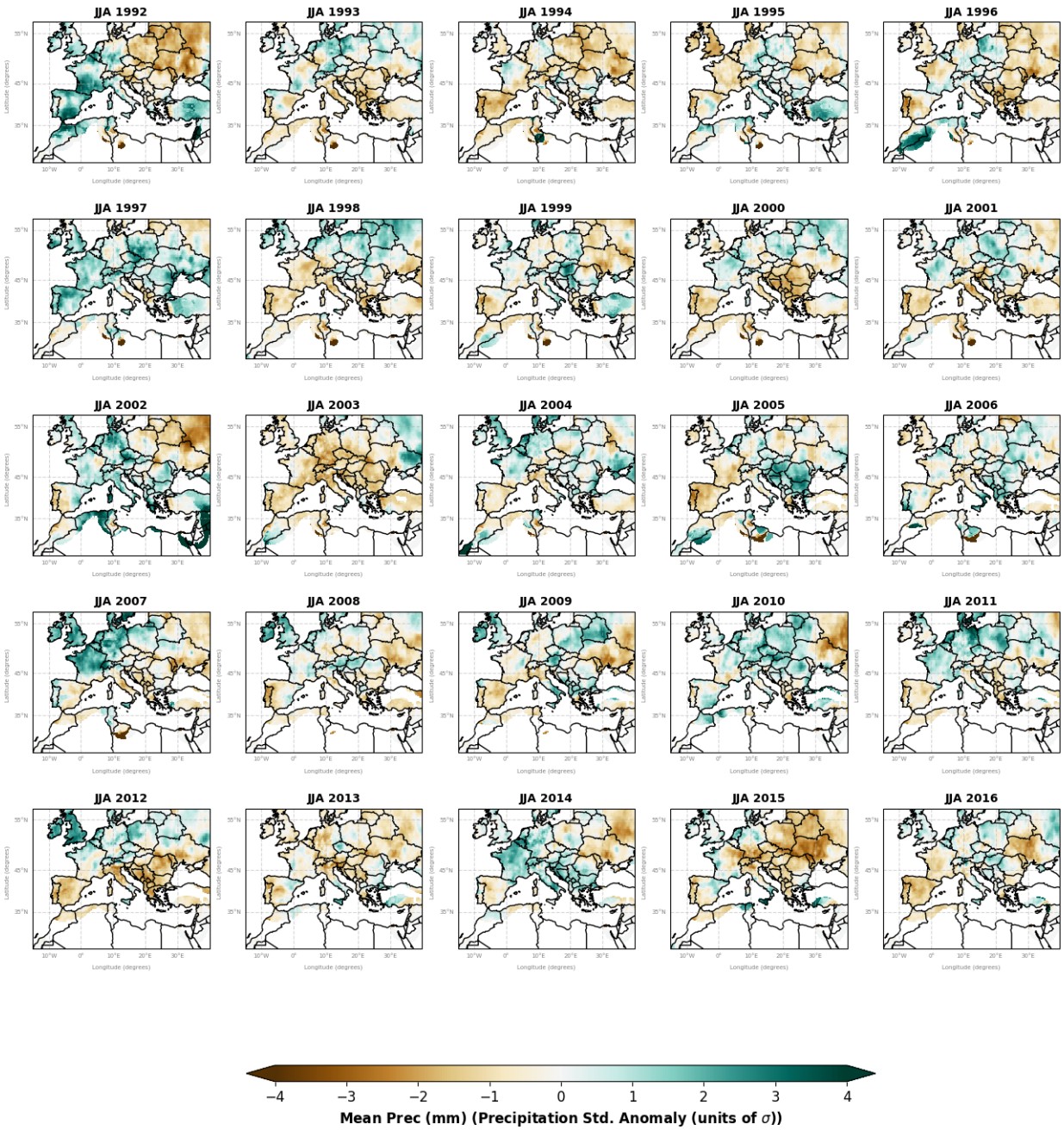

**Figure S.1.** Precipitation standardized anomalies (units: standard deviation, $\sigma$) computed over the summers (June-July-August, JJA) between 1970 and 2016 using the E-OBSv20e data-set and referring to the 1970–1990 precipitation and temperature climatology.

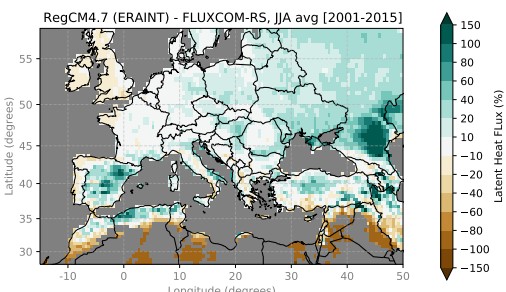

**Figure S.2.** Spatial distribution of summer percentage biases (units: %) in latent heat fluxes between the RegCM4-CLM4.5 model and the FLUXCOM data-set (Remote-Sensed product) over the period 2001–2015. For comparison, model output have been remapped onto the FLUXCOM grid.

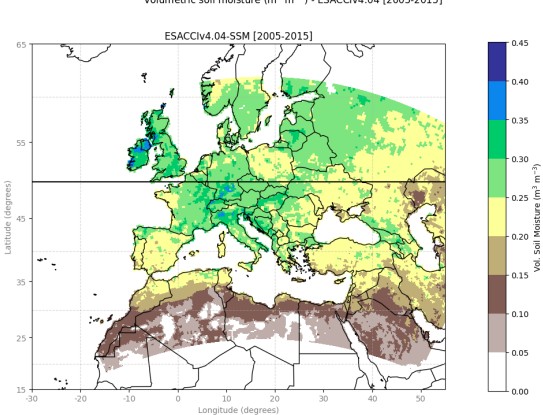

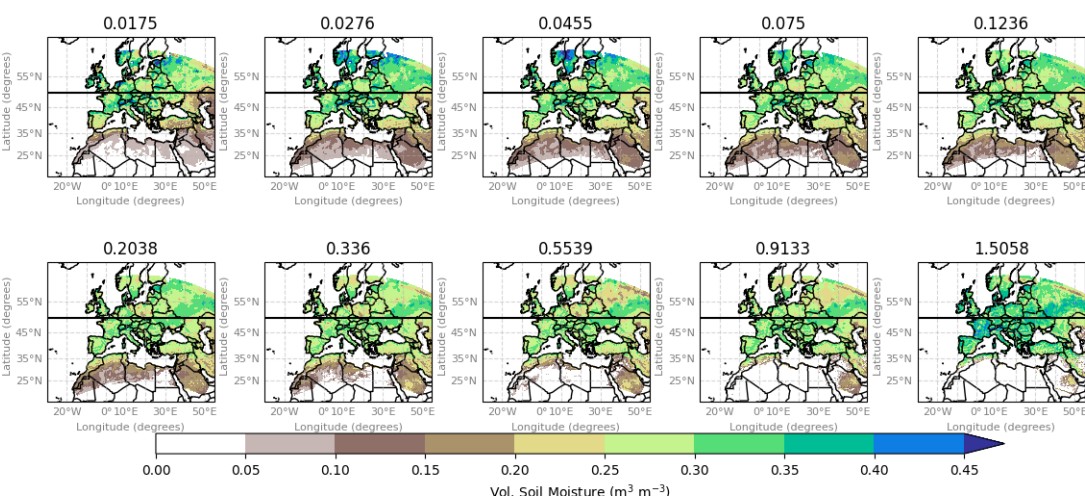

**Figure S.3.** Comparison of volumetric soil moisture ($m^3$ $m^{-3}$) between the ESACCIv4.04 data-set and the RegCM4-CLM4.5 model over the period 2005–2015. For comparison, model output was remapped onto the ESACCI grid.

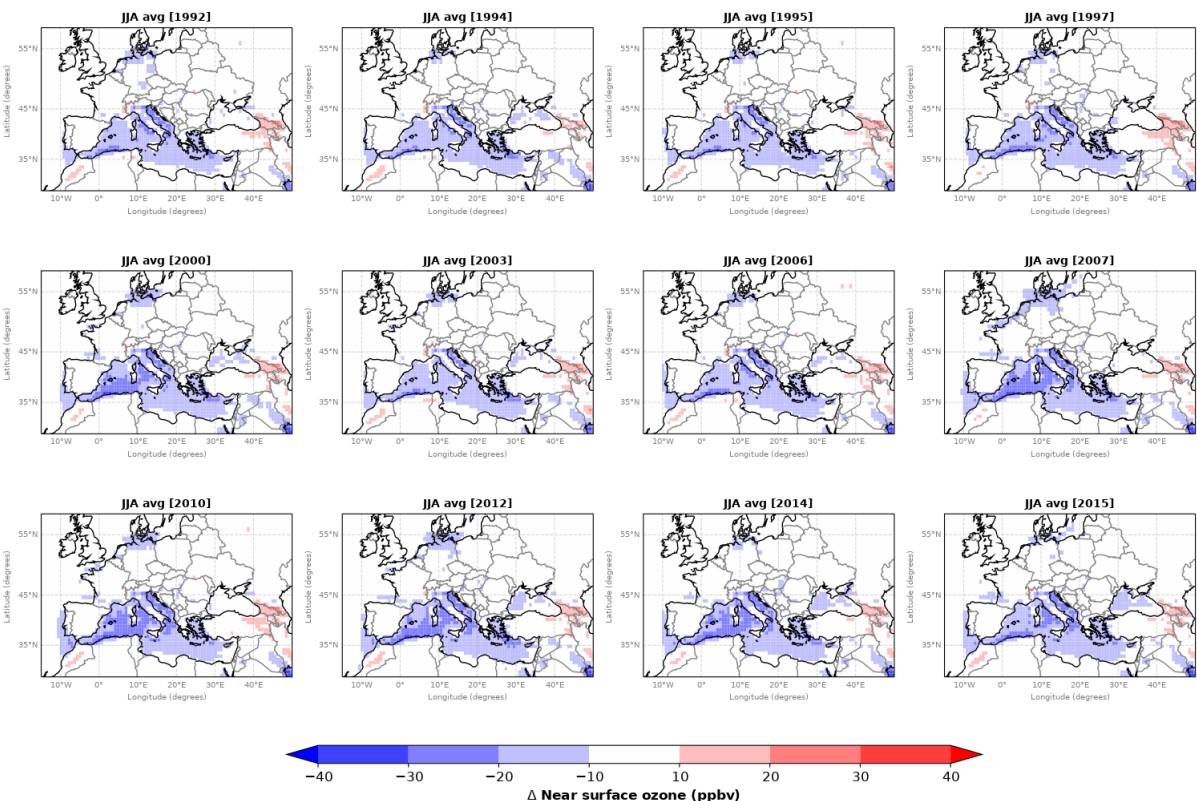

**Figure S.4.** Spatial distribution of summer-averaged differences in ozone (O3) volume mixing ratio at 1000 hPa (units: ppbv) between the RegCM4-chem model and the CAMS re-analyses.

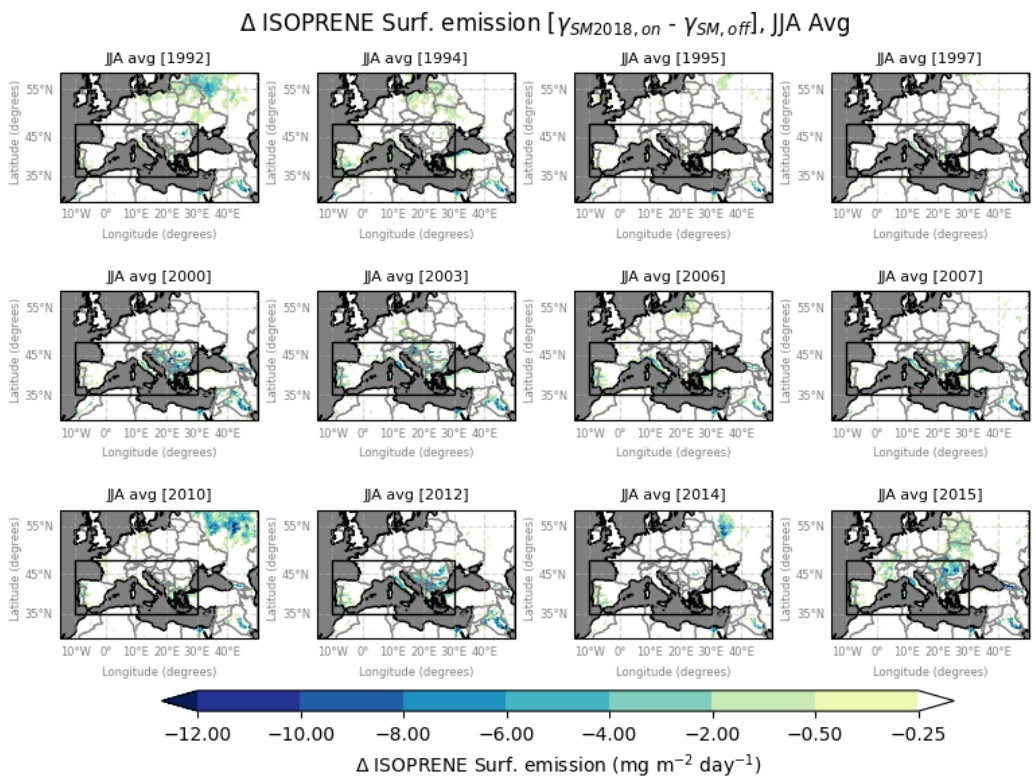

**Figure S.5.** Spatial distribution of summer-averaged absolute changes in isoprene emissions (units: $\mathrm{mg\,m^{-2}\,day^{-1}}$) as simulated by the RegCM4chem-CLM4.5-MEGAN2.1 model across the selected summers over the period 1992–2016. Absolute changes were computed as the difference between summer averages (JJA) of model output from the GAMMA-SM2018on and the GAMMA-SMoff simulations. Black boxes highlight the Euro-Mediterranean region selected for analysis.

## JJA Standardized Anomaly, E-OBSv20ens Mean Temp (1970-2016 vs. 1970-1990), Res. 0.25°

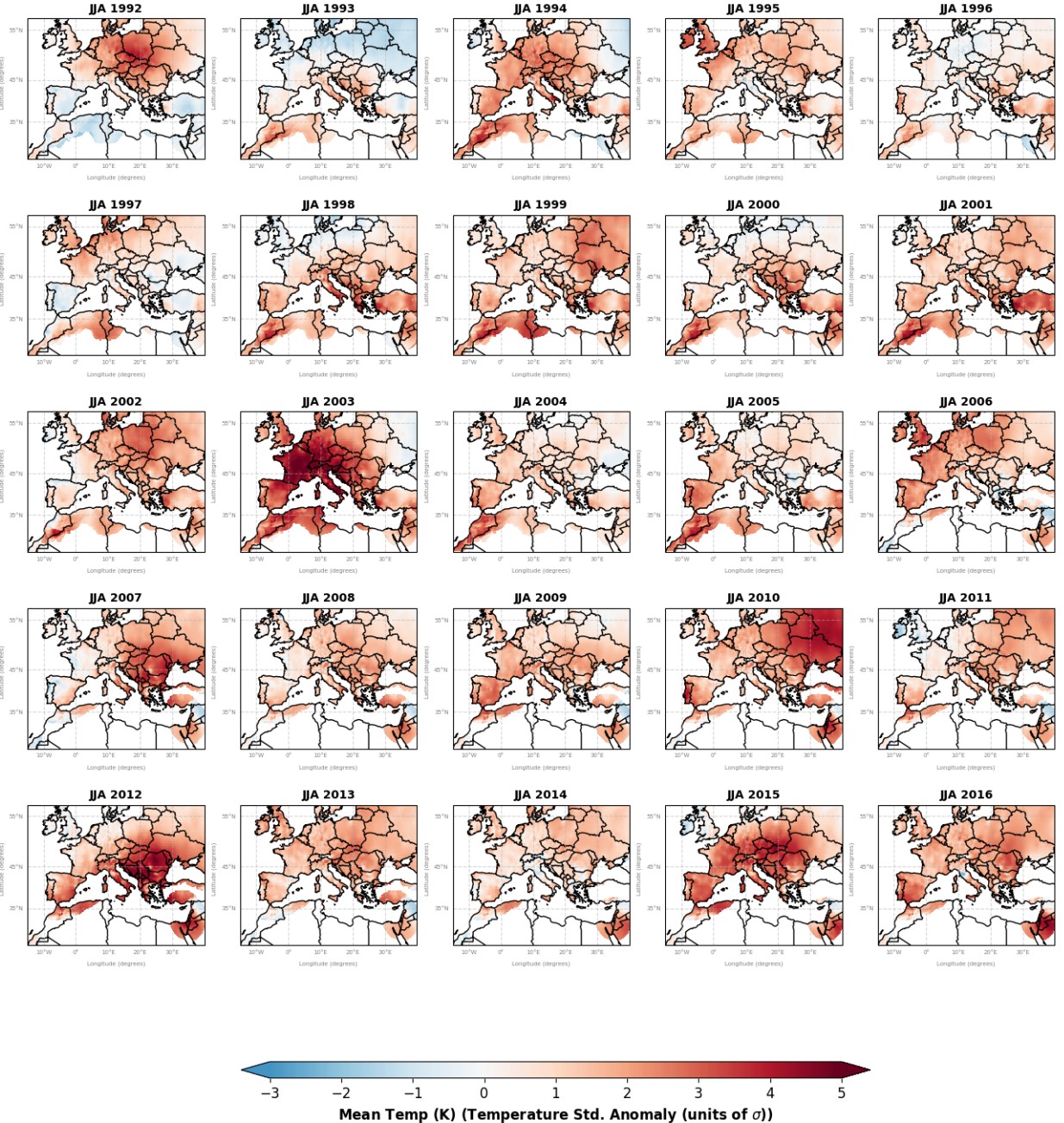

**Figure S.6.** Standardized anomalies (units: standard deviation, $\sigma$) in mean surface air temperatures computed over the summers between 1970 and 2016 using the E-OBSv20e data-set and referring to the 1970–1990 precipitation and temperature climatology.

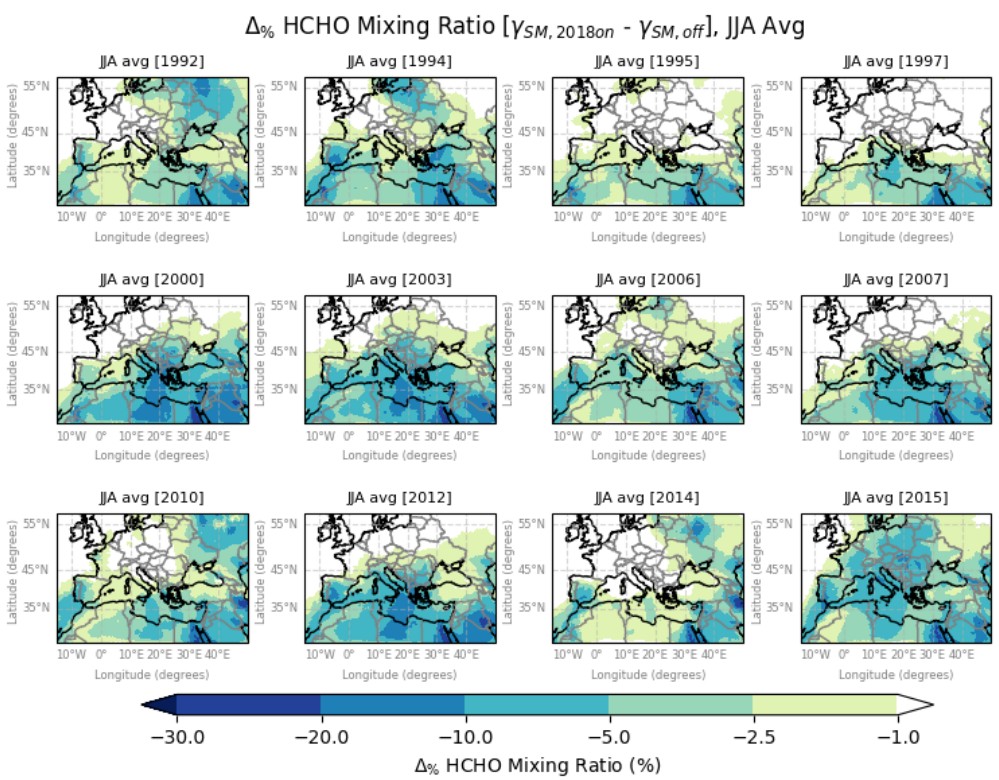

**Figure S.7.** Spatial distribution of summer-averaged percentage changes in formaldehyde surface mixing ratio (units: %) as simulated by the RegCM4chem-CLM4.5-MEGAN2.1 model across the selected summers over the period 1992–2016. To compute percentage changes, the difference between summer averages from the GAMMA-SM2018on and the GAMMA-SMoff simulations was divided by the reference simulation, GAMMA-SMoff.

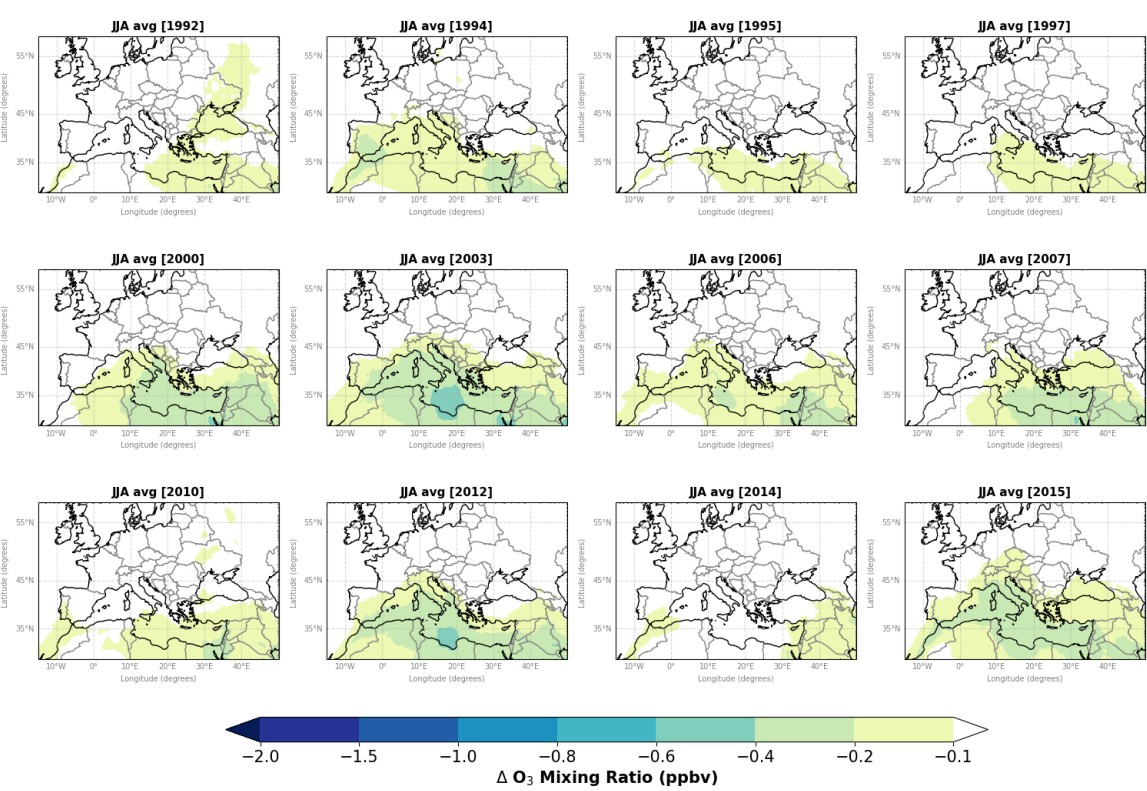

**Figure S.8.** As Figure S.7, spatial distribution of absolute changes in ozone mixing ratio at 1000 hPa (units: ppbv).

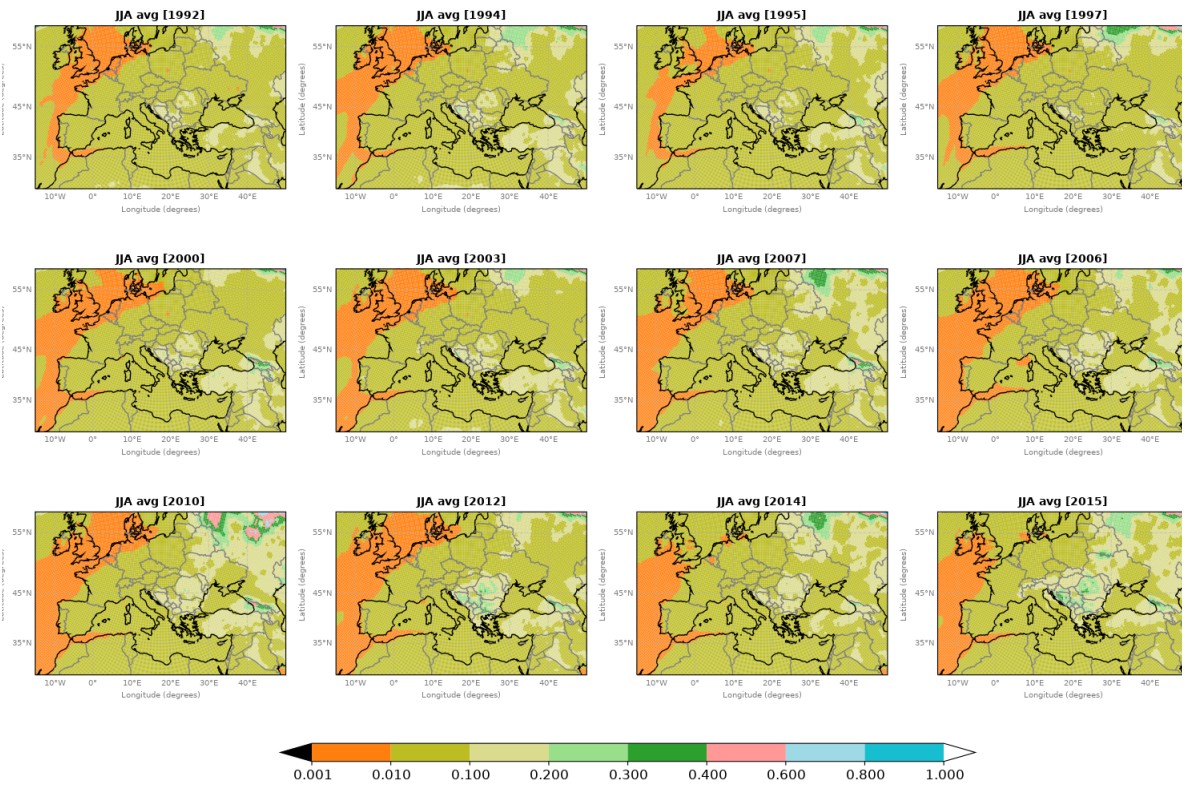

**Figure S.9.** Spatial distribution of the ratio between formaldehyde (HCHO) and nitrogen di-oxide (NO2) mass mixing ratios at 1000 hPa as simulated by the RegCM4-chem model. The HCHO/NO2 ratio results lower than 1 over the whole domain, indicating that the model reproduces a VOC-limited regime, based on Duncan et al. (2010).