# Peer review of "Assessment of isoprene and near surface ozone sensitivities to water stress over the Euro-Mediterranean region"

_EGUsphere, 2022_

## Author Response (AR1)

**Assessment of isoprene and near surface ozone sensitivities to water stress over the Euro-Mediterranean region**

Strada, S.[1], Pozzer, A., Giorgi, F., Giuliani, G., Coppola, E., Solmon, F., Jiang, X., Guenther, A., Bourtsoukidis, E., Serça, D.

**Responses to Reviewers**
* * *
[1]*Corresponding author*: sstrada@ictp.it

**Ms. Ref. No.**: egusphere-2022-1522

**Title**: Assessment of isoprene and near surface ozone sensitivities to water stress over the Euro-Mediterranean region

**Authors**: Strada et al.

**Journal**: Atmospheric Chemistry and Physics

**Response to Reviewer #1**

GENERAL COMMENT:

This is a detailed study of how emission of isoprene are impacted by the newer MEGAN v3 soil moisture activity parameterisation compared to the previous v2.1 parameterisation. Comparisons are also made using satellite formaldehyde columns and surface ozone measurements. Results were compared over a number of summer seasons from 1992 to 2015, when isoprene is expected to be at peak concentrations in Europe.

One of the interesting parts of this paper was the demonstration of how 'smooth' the soil moisture activity function is using the MEGANv2.1 parameterisation compared to MEGANv3. The latter parameterisation allowed for more spatially varying reductions in isoprene, which were often less than those calculated using MEGANv2.1 and more localised. By contrast, a soil moisture activity function of ~0.4-0.5 covers most of Europe in summer using MEGANv2.1 which causes a very even, but perhaps too high a reduction in isoprene emissions.

I thought the methods, model and observations section was very well detailed, with all datasets well documented and described.

I only have a few comments before publication is recommended. They mainly relate to difficulties reading the figures.

**Authors' response**: We thank Reviewer #1 for this positive evaluation of our article and these insightful comments. Below, we separately reply to each comment and, when necessary, we precise where and how the manuscript has been modified. While reviewing the manuscript, we found an error in the conversion of ozone and formaldehyde mixing ratios that has been corrected in the revised manuscript.

SPECIFIC COMMENTS

**Line 139**: Would be good to have these values tabulated somewhere, or even refer to Oleson et al (2013) table 8.1 which is where I finally found them. Would be useful if others wanted to implement the new scheme.

**Authors' response**: Thanks to this comment, we have added the reference to Oleson et al. (2013) in Table 8.3 in which the plant functional type (PFT) root distribution parameters are listed (Table 8.1 has PFT photosynthetic parameters). The text has changed, now reading:

"The root fraction distribution $r$ decreases exponentially with depth based on PFT-dependent parameters (see Table 8.3 in Oleson et al., 2013)."

**Line 343**: 76 mg/m2/day is a huge reduction. I wondered where about this was located, and what was the underlying vegetation type?

**Authors' response**: Thanks to this insightful suggestion, we found that isoprene emissions have decreased by -76 mg/m$^2$/day in July 2010 over a grid-cell located in south-western Russia (latitude: 60.24°N; longitude: 39.88°E). Over this grid cell, based on the CLM4.5 land cover, the boreal needleaf evergreen (49%) and the temperate broadleaf deciduous (47%) trees dominate, with the remaining 4% covered by crop (C3 unmanaged rainfed crop). Based on this information, we modified the manuscript accordingly:

"In the summer 2010, the RegCM4chem-CLM4.5-MEGAN2.1 model also reproduces the largest decrease in isoprene emissions, with a maximum reduction of -76 mg m$^{-2}$ day$^{-1}$ simulated in July and located over south-western Russia (latitude: 60.24°N; longitude: 39.88°E) where needleaf evergreen and broadleaf deciduous trees dominate in the CLM4.5 land cover. Between July and August 2010, south-western Russia was hit by an extreme heat wave and drought (Barriopedro et al., 2011). Such a substantial reduction in isoprene emissions corresponds to -3 mg m$^{-2}$ hour$^{-1}$ (not shown)."

**Line 481**: there looks to be a co-author comment (?) still in the text.

**Authors' response**: The question marks is actually a reference that did not work during the compilation of the Latex source. We have now correctly inserted the reference to the study by Massad et al. (2019). The correct text is :

"Moreover, the modelling of ozone chemistry strongly depends on the spatial resolution that influences the model ability in adequately distinguish chemical regimes (i.e., VOC- or NOx- limited) that, in turn, depend on the emission pattern of natural and anthropogenic sources (Massad et al., 2019)."

**Conclusions section**. There are a lot of new references introduced here which isn't usual – they're more suited to the introduction where previous literature is more commonly reviewed.

**Authors' response**: Based on this suggestion, we have restructured the Introduction and the Conclusion sections. Now, previous literature is mainly reviewed in Sect. 1, while Sect. 4 contains references related to recommendations for future studies.

**Figures**: Most were too small to see properly.

**Authors' response**: We improved all figures and made them bigger. We also implemented colorblind-friendly colormaps in Figures 6 (now Figure 7) and 11 (now Figure 12), as requested by the EGU journals.

**Figure 1** the axis text is too feint to read.

**Authors' response**: We increased the size of Fig. 1, and as well of colorbar labels and ticks.

**Figure 4** needs the units putting on the y-axis. The orange line is also too feint to see.

**Authors' response**: In Fig. 4, we added the units on the Y-axis, while in Fig. 5 we thickened the orange line.

**Figure 5**: I was confused by the legend which has pointers indicating the scale goes below 0 and above 1. There is a lot of white areas in the 12 maps which look to be above 1 and suggests that gamma2018 is higher than the default (which it can't be)?

**Authors' response**: The Reviewer is correct: the soil moisture activity factor $\gamma_{SM}$ ranges between 0 (water stress shuts down isoprene emissions) and 1 (there is no water stress). To produce all figures, we use Python which does not include the upper limit value of each color bin. For this reason, grid-cells where $\gamma_{SM}$ equals 1 appear as white areas in Figure 6 (now Figure 7, while Figure 5 shows the model evaluation for formaldehyde column concentrations). To solve this issue, we forced Python to include grid-cells where $\gamma_{SM}$ equals 1 in the last colored bin (dark blue). Using the same method, we also modified Figure 11 (now Figure 12), which compares the two soil moisture activity factors. In the new figures, there are no grid-cells colored in white. In addition, we implemented colorblind-friendly colormaps, as requested by the EGU journals.

**Figure 9a**: numbers on y-axis are bunched together and overlap

**Authors' response**: To ease the reading, we modified the angle rotation of the Y-axis labels.

**Ms. Ref. No.**: egusphere-2022-1522

**Title**: Assessment of isoprene and near surface ozone sensitivities to water stress over the Euro-Mediterranean region

**Authors**: Strada et al.

**Journal**: Atmospheric Chemistry and Physics

**Response to Reviewer #3**

GENERAL COMMENT:

This study applies a regional vegetation-climate-chemistry model to investigate the influence of water stress on isoprene emissions and surface ozone over Europe in 1992-2016. This is done by coupling the land module and biogenic emission module to derive the soil water stress function, which is then used to determine soil moisture activity factor in the parameterization of isoprene emission. Simulation results show that water stress reduces summertime isoprene emissions on average by nearly 6%, and by -20 to -60% in extreme dry summers, but influence on ozone is relatively small. This study is well-designed, easy to follow, and the results are useful for the community. It has room to be improved by addressing the following comments.

**Authors' response**: We thank Reviewer #3 for this positive evaluation of the study and for the insightful comments. Below, we separately reply to each comment and, when necessary, we precise where and how the manuscript has been modified. Supplementary figures produced to answer to reviewers' comments have been gathered in a separated document ("Responses_to_Reviewers_fig.pdf"). To avoid confusion with figures in the manuscript and in the Supplementary Material, we customized the counter as Figure R.*. While reviewing the manuscript, we found an error in the conversion of ozone and formaldehyde mixing ratios that has been corrected in the revised manuscript.

SPECIFIC COMMENTS

1. My major concern is that this study lacks direct evaluation with observed isoprene emissions. Does the inclusion of water stress effect improve the simulation of isoprene emissions? Does the new scheme outperform the old scheme?

**Authors' response**: We agree with the Reviewer that direct evaluation with observed isoprene emissions would be the best choice to assess the model performance and compare the old and new schemes that link the effect of soil moisture on isoprene emissions. However, since there is no network over Europe routinely measuring isoprene emissions, or isoprene concentrations, in vegetated areas, we focused the model evaluation on a proxy of isoprene emissions such as formaldehyde (HCHO). Nevertheless, we have now also a limited evaluation of isoprene concentrations as simulated by the RegCM4chem-CLM4.5-MEGAN2.1 model in the GAMMA-SMoff simulation. The model output have been compared against observations collected during two field campaigns. Figure R.1 shows the comparison against isoprene concentrations measured in south-eastern France (site: La Verdière; Latitude: 43.63° N, Longitude: 5.93° E) during the summer 2000 (from June 21 to July 6) in the framework of the ESCOMPTE field campaign (Cros et al., 2004) when isoprene concentrations had been measured every 30 minutes using a Fast Isoprene Sensor. In Figure R.2, model output have been compared against data collected in Cyprus (site: Ineia; Latitude: 34.96° N, Longitude: 32.39° E)

during the summer 2014 (from July 7 to August 3; data collected nearly every 30 minutes) using techniques of gas chromatography - mass spectrometry.

At both sites, RegCM underestimates isoprene concentrations, which is consistent with RegCM underestimating concentrations of a proxy of isoprene such as formaldehyde, as shown in Section 3.1.3 in the manuscript. Sometimes, the model reproduces a delayed peak in isoprene concentrations compared to observations. Differences between observations and model output could result from multiple factors, for example:

1.    The cold and wet model bias (see Fig. 1 in the manuscript) that limits isoprene emissions;

2.    Differences between the dominant vegetation types on the field and in the model grid-cell. For example, in La Verdière, vegetation is mainly characterized by Mediterranean oak forest (more than 80% of *Quercus Pubescens,* which is a deciduous tree), while the RegCM grid-cell is mainly covered with needle-leaf evergreen temperate trees (36%), and C3 grass (37%), which are both low isoprene emitters, and has only a small amount of broadleaf deciduous trees (6%).

3.    Different scales: a model grid-cell spans over a surface of around 25x25 km$^2$, while station measurements have a footprint of a few hundreds of meters, depending on the terrain where the observations have been collected. For example, Ineia is located close to the seacoast leading to use a model grid-cell that is located over the sea.

Based on these results, we have added the comparison between model output and in-situ measurements of isoprene concentrations in Section 3.1.3 in the revised manuscript.

Cros, B., et al.: The ESCOMPTE Program: an overview. Atmospheric Research, 2004, 69 (3-4), pp.241-279, DOI: 10.1016/j.atmosres.2003.05.001_x005F_xffff_.

2. I also wonder how the water stress effect changes the relationship between isoprene emissions and temperature. Previous studies have revealed the decrease in isoprene emission in extreme high temperature, but would the water stress effect further change the turning point of the T-emission curve? Some discussions would be useful.

**Authors' response**: This is an interesting question.

The version of the MEGAN model (2.1) implemented in the RegCM4.7 model accounts for the effect of past temperatures on isoprene emissions (over the past ten days): the warmer the temperatures, the higher the emissions (see Fig. 4 in Guenther et al., 2006). However, this version does not account for the decrease in isoprene emissions due to extreme high temperatures.

Extreme high temperatures often co-occur with droughts. It is, then, not trivial to separate the effect of these climate extremes on isoprene emissions. In our simulations, we observed the largest decreases in isoprene emissions in the summers 2003 and 2012 (see Table 4) when the observation-based summer temperatures are nearly 4–5 standard deviations above (warmer than) the 1970–1990 climatology, as shown by Figure S.6 in the Supplementary Material. These results and those mentioned by the Reviewer, which reveal the decrease in isoprene emissions under extreme high temperatures, suggest that isoprene emissions would be strongly reduced when heat wave and drought co-occur. In the revised manuscript, we added the following comment in Sect. 3.2:

"The largest decreases in isoprene emissions occur in the summers 2003 and 2012 (Table 4), when the observation-based summer temperatures are nearly 4–5 standard deviations above (warmer than) the 1970–1990 climatology (Fig. S.6). These results suggest that isoprene emissions would be strongly reduced when heat wave and drought co-occur."

3. Related to point 2, I think some discussions on the extreme ozone episodes would be helpful to evaluate the effect of water stress on ozone concentration.

**Authors' response**: Extreme high temperatures often exacerbate ozone pollution and lead to extreme ozone episodes. In a second paper (in preparation), we investigated the ozone climate penalty by performing simulations under both present-day (1990–2004) and future climates (2035–2049). To assess the impact on ozone concentration of both the direct effect of high temperatures and the indirect effect of water stress on isoprene emissions, we designed two sensitivity simulations over the summer 2003 with and without the soil moisture activity factor activated. In these simulations, we artificially increased air temperature across the whole atmospheric column by a fixed amount varying with atmospheric levels. Results show that an average increase of 2.5°C in air temperature leads to an increase in surface-ozone level smaller than 2 ppbv (1%), regardless the soil moisture activity factor is activated or not. Although the paper will be submitted soon, we added the following comment in Sect. 4:

"In a future study, we aim to explore the ozone climate penalty over the Euro-Mediterranean region under both present-day and future climates and to assess the impact on ozone concentration of both the direct effect of high temperatures and the indirect effect of water stress on isoprene emissions."

4. I feel that the model evaluation of chemical fields is rather insufficient. It only shows the mean magnitude of observed and simulated HCHO column and ozone concentrations. How well does the model capture the spatial and temporal pattern of HCHO and ozone? It might be also important to evaluate the ozone chemical regime (NOx-limited or VOCs-limited) somewhere.

**Authors' response**: We agree with the Reviewer that the model evaluation for chemical species could be extended. In our study, we focused on the analysis of differences between sensitivity simulations, therefore possible systematic biases in the model would cancel out, thus giving robust results. However, we performed some additional evaluation for both ozone and formaldehyde by comparing model output against re-analyses from the Copernicus Atmosphere Monitoring Service (CAMS: Marecal et al., 2015) for the period 2003–2007. In addition, surface ozone concentrations have been evaluated against observations collected in La Verdière (France) in the summer 2000 (from June 21 to July 6) using an conventional UV absorption ozone analyzer (Environement S.A., Poissy, France, model O3 41M). Ozone concentrations have been measured during the ESCOMPTE field campaign, together with isoprene measurements shown in Figure R.1.

For near surface ozone, model output are lower than CAMS re-analyses with differences between 10 and 20 ppbv over the Mediterranean Basin, with some summers and few grid-cells showing differences between 20 and 30 ppbv (Fig. R.3). The model underestimates near-surface ozone as well when compared to in-situ measurements (Fig. R.4); in particular, model output shows a smaller variability than in-situ measurements. For formaldehyde, model outputs near the surface are also lower than CAMS reanalyses with differences between -1 and -4 ppbv (Fig. R.5).

We also assessed the ozone chemical regime using the ratio between formaldehyde (HCHO) and nitrogen dioxide ($NO_2$) as presented in Duncan et al. (2010). Results show that the model reproduces a VOC-limited regime over the whole domain, with a $HCHO/NO_2$ ratio lower than 1 (Fig. R.6). Based on this analysis, in the revised manuscript we updated all discussions about the ozone regime over the model domain.

Duncan, B. N. et al.: *Application of OMI observations to a space-based indicator of NOx and VOC controls on surface ozone formation*, Atmospheric Environment, Volume 44, Issue 18, 2010, Pages 2213-2223, DOI: https://doi.org/10.1016/j.atmosenv.2010.03.010.

Marécal, V. et al.: *A regional air quality forecasting system over Europe: the MACC-II daily ensemble production*, Geoscientific Model Development, 8, 2777–2813,750, https://doi.org/10.5194/gmd-8-2777-2015, 2015.

5. The figure quality can be improved. For example, the size of the figure is often too small compared to the colorbar (Figs 11 and 12), and in some cases the label is missing (Fig.5).

**Authors' response**: We improved all figures. We added axis labels to Figure 5 (now Fig. 6), we increased the size of figures compared to the colorbars, and we implemented colorblind-friendly colormaps in Figures 6 (now Fig. 7) and 11 (now Fig. 12), as requested by the EGU journals.

[Figure]

Figure R.1: Comparison of the time-series of isoprene concentrations (units: pptv) collected at La Verdière (Latitude: 43.63° N; Longitude: 5.93° E; France) during the ESCOMPTE filed campaign performed in the summer 2000. The green solid line shows observations, while the blue solid line shows the model output extracted over the nearest grid-cell to the observation spot from the GAMMA-SMoff simulation performed with the RegCM4chem-CLM4.5-MEGAN2.1 model.

[Figure]

Figure R.2: Comparison of the time-series of isoprene concentrations (units: pptv) collected at Ineia (Latitude: 34.96° N; Longitude: 32.39° E; Cyprus) during the summer 2014. The green solid line shows observations, while the blue solid line shows the model output extracted over the nearest grid-cell to the observation spot from the GAMMA-SMoff simulation performed with the RegCM4chem-CLM4.5-MEGAN2.1 model.

[Figure]

Figure R.3: Spatial distribution of summer-averaged differences in ozone ($O_3$) volume mixing ratio at 1000 hPa (units: ppbv) between the RegCM4-chem model and the CAMS re-analyses.

[Figure]

Figure R.4: Comparison of the time-series of near-surface ozone concentrations (units: ppbv) collected at La Verdière (Latitude: 43.63° N; Longitude: 5.93° E; France) during the ESCOMPTE filed campaign performed in the summer 2000. The green solid line shows observations, while the blue solid line shows the model output extracted over the nearest grid-cell to the observation spot from the GAMMA-SMoff simulation performed with the RegCM4chem-CLM4.5-MEGAN2.1 model.

[Figure]

Figure R.5: Spatial distribution of summer-averaged differences in formaldehyde (HCHO) volume mixing ratio at 1000 hPa (units: ppbv) between the RegCM4-chem model and the CAMS re-analyses.

[Figure]

Figure R.6: Spatial distribution of the ratio between formaldehyde (HCHO) and nitrogen di-oxide (NO$_2$) mass mixing ratios at 1000 hPa as simulated by the RegCM4-chem model. The HCHO/NO$_2$ ratio results lower than 1 over the whole domain, indicating that the model reproduces a VOC-limited regime, based on Duncan et al. (2000).